# Effect of Polyoxymethylene (POM-H Delrin) offgassing within Pandora head sensor on direct sun and multi-axis formaldehyde column measurements in 2016 - 2019

Elena Spinei[1], Martin Tiefengraber[2,3], Moritz Müller[2,3], Manuel Gebetsberger[2], Alexander Cede[2], Luke Valin[4], James Szykman[4], Andrew Whitehill[4], Alexander Kostakis[5], Fernando Santos[6], Nader Abbuhasan[7], Xiaoyi Zhao[8], Vitali Fioletov[8], Sum Chi Lee[8], and Robert Swap[9]

[1]Center for Space Science And Engineering Research, Virginia Polytechnic Institute and State University, Blacksburg, VA, USA
[2]LuftBlick, Innsbruck, Austria
[3]Department of Atmospheric and Cryospheric Sciences, University of Innsbruck, Innsbruck, Austria
[4]United States Environmental Protection Agency, Durham, NC, USA
[5]Universities Space Research Association, Columbia, MD, USA
[6]Earth System Science Interdisciplinary Center, University of Maryland, College Park, MD, USA
[7]Joint Center for Earth Systems Technology, University of Maryland, Baltimore County, Baltimore, USA
[8]Air Quality Research Division, Environment and Climate Change Canada, Toronto, M3H 5T4, Canada
[9]NASA Goddard Space Flight Center, Greenbelt, MD, USA

**Correspondence:** Elena Spinei: eslind@vt.edu

**Abstract.**

Analysis of formaldehyde measurements by the Pandora spectrometer systems between 2016 and 2019 suggested that there was a temperature dependent process inside Pandora head sensor that emitted formaldehyde. Some parts in the head sensor were manufactured from thermal plastic polyoxymethylene homopolimer (E.I. Du Pont de Nemour & Co., USA: POM-H Delrin®)

and were responsible for formaldehyde production. Laboratory analysis of the four Pandora head sensors showed that internal formaldehyde production had exponential temperature dependence with a damping coefficient of $0.0911 \pm 0.0024 \, °C^{-1}$ and the exponential function amplitude ranging from 0.0041 DU to 0.049 DU. No apparent dependency on the head sensor age and heating/cooling rates was detected.

The total amount of formaldehyde internally generated by the POM-H Delrin components and contributing to the direct sun

measurements were estimated based on the head sensor temperature and solar zenith angle of the measurements. Measurements in winter, during colder (<10°C) days in general and at high solar zenith angles (> 75 °) were minimally impacted. Measurements during hot days (>28°C) and small solar zenith angles had up to 1 DU ($2.69 \, \mathrm{molecules/cm^2}$) contribution from POM-H Delrin parts. Multi-axis differential slant column densities were minimally impacted (< 0.01 DU) due to the reference spectrum collected within a short time period with a small difference in head sensor temperature.

Three new POM-H Delrin free Pandora head sensors (manufactured in summer 2019) were evaluated for temperature dependent attenuation across the entire spectral range (300 to 530 nm). No formaldehyde or any other absorption above the instrumental noise was observed across the entire spectral range.

## 1   Introduction

The importance of formaldehyde (HCHO) in tropospheric chemistry arises from its participation in radical formation and recycling, including $HO_x$ ($HO + HO_2$) (Liu et al., 2007; Alicke, 2002). The $HO_x$ budget, in its tern, determines the oxidation capacity of the atmosphere and formation of photochemical smog ($O_3$) in the lower troposphere in the presence of $NO_x$ and sunlight. Since the major path of HCHO into the atmosphere is through oxidation of non-methane volatile organic compounds (NMVOC) and daylight removal is mostly through the photolysis and oxidation by HO, daytime HCHO abundances above

background levels are mainly indicative of local emissions and the local oxidizing capacity of the atmosphere.

The most efficient way to monitor geographical and temporal variability of HCHO on a global scale is from satellite platforms. Satellite observations of HCHO from sun synchronous polar orbits started with GOME in 1995. Since then several instruments provided global coverage with various spatial and temporal resolution (Zhu et al. (2016a), Wang et al. (2017): SCIAMACHY (10:00 hr LT, 32x215 $km^2$, 2002 - 2012, Wittrock et al., 2006, De Smedt et al. (2008)); GOME-2A and -2B

(09:30 hr LT, 40x40 $km^2$, since 2007 to the date of publication Pinardi et al. (2020), De Smedt et al. (2015), De Smedt et al. (2012), Hewson et al. (2013)), OMI (13:30 hr LT, 13x24 $km^2$, since 2004 to the date of publication, Herman et al. (2018), Pinardi et al. (2020), De Smedt et al. (2015)); TROPOMI (13:30 hr LT, 3.5x7 $km^2$, since 2018 to the date of publication, Verhoelst et al. (2020), De Smedt et al. (2018), Vigouroux et al. (2018)). Next generation air quality instruments, positioned in geostationary orbit, will provide unprecedented temporal coverage over Asia (GEMS, since February 18, 2020, Kwon et al.

(2019)); North America (TEMPO, estimated launch in early 2022); Europe (Sentinel-4, estimated launch in 2023).

Current and future satellite HCHO observations require routine and systematic validation through the use of independent measurements to assess biases and uncertainties and encourage full utilization of satellite data to support both science and applications. Validation of satellite HCHO products, however, is challenging due to spatial and temporal sampling differences among the satellite, ground-based (e.g. FTIR (Vigouroux et al., 2009), DOAS (Chan et al., 2020)) and airborne platforms (Zhu

et al., 2016b). Ideally, it involves data from ground-based networks of identical instrumentation with continuous measurements and uniform data analysis and wide global distribution. To meet current and future satellite validation needs (e.g. TEMPO, TROPOMI) ground-based HCHO column measurements should have an accuracy better than 0.1 DU (1 DU = $2.69 \cdot 10^{16}$ molecules/$cm^2$, expected nominal TEMPO precision over 3 hr is $1.95 \cdot 10^{15}$ molecules/$cm^2$).

Pandonia Global Network (PGN) is a NASA and ESA sponsored ground-based network of standardized and homogeneously

calibrated Pandora instruments focused on air quality and atmospheric composition measurements. The main objective of PGN is to provide systematic data processing and data dissemination to the greater global community in support of in situ and remotely sensed AQ monitoring (Szykman et al., 2019). One of the PGN's major objectives is the validation of satellite-based UV-visible sensors, specifically, Sentinel 5P, TEMPO, GEMS and Sentinel 4. PGN is focused on providing measurements of the total column and vertically resolved concentrations of a range of trace gases (e.g., $NO_2, O_3, HCHO, SO_2$). Pandora total $NO_2$

column measurements have been extensively used for OMI validation (Herman et al. (2009); Pinardi et al. (2020); Herman

et al. (2019); Verhoelst et al. (2020)) and atmospheric composition studies during multi-agency supported field campaigns such as DISCOVER-AQ (Reed et al., 2015), KORUS-AQ (Spinei et al., 2018), OWLETS (Gronoff et al., 2019)), and LISTOS (https://www-air.larc.nasa.gov/missions/listos).

The Pandora spectrometer system, deployed within PGN, is a cost-effective ground-based instrument, operating on the principle of the passive UV-visible differential optical absorption spectroscopy technique (DOAS). Pandoras undergo extensive laboratory characterization and have a robust data acquisition and analysis software package, Blick Software Suite (Cede, 2019). Pandora instruments are fully automated and fully programmable to perform all types of DOAS observation geometries (e.g., direct sun, multi-axis, and target) from sunrise to sunset and overnight for moon measurements. Pandoras have no consumables and are designed for unattended operation in outdoor environments. Measured spectra are automatically collected and submitted to the PGN server via an Internet connection for centralized uniform real-time processing by the Blick Software Suite. Pandora instrument consists of a small Avantes low stray light spectrometer ($280 - 530$ nm with 0.6 nm full width at half maximum spectral resolution) connected to a telescope assembly by a 400-micron core diameter single strand fiber optic cable. The telescope assembly (head sensor) is attached to a 2-axis positioner (SciGlob), capable of accurate pointing ($\pm 0.1°$). A diffuser is included in the optical path to minimize the effect of small pointing errors for direct sun measurements with a 2.5° full width half maxima (FWHM) field of view (FOV). Pandoras measure scattered solar photons without the diffuser with 1.5° FWHM FOV.

Here we present (a) the discovery of the Pandora instrument artifact due to POM-H Delrin plastic offgassing impacting the Pandora HCHO measurements up to 2019 (Section 2); (b) a laboratory and field characterization of the interference on direct sun and multi-axis results (Sections 3, 4, 5); (c) characterization of the interference following an engineered solution (Sections 3, 4). Results presented in this study show that plastic related HCHO offgasing significantly impacted direct sun total columns and minimally impacted multi-axis retrievals (Section 5). Due to strong temperature dependence of HCHO offgasing the largest interference was observed in summer. Extensive analysis of the Pandora instruments, after the engineering solution was implemented, shows no interference and strengthens confidence in future direct sun measurements. Table 1 lists the Pandora instrument description and contribution to this study.

## 2 Pandora HCHO measurements

Pandora instruments were first field deployed in 2006 with the main focus on direct sun $O_3$ and $NO_2$ total columns measurements. Retrieval of weak absorbers such as HCHO was not possible from the pre-2016 Pandora direct sun measurements due to the telescope assembly front window etaloning. Window introduced interference was larger than the background HCHO absorption level and is not correctable in the pre-2016 measurements (Park et al., 2018).

Pandora instrumentation has undergone several design changes that significantly improved HCHO direct sun measurements (e.g. 64 row CCD, new tracker). In spring 2016, the telescope assembly front window was replaced with a window containing an anti-reflective coating (ARC). This reduced the etaloning interference and improved the ability to retrieve formaldehyde

**Table 1.** Pandora instruments used in this study

| N | Owner | Manufactured | Relevant Hardware Components | Contribution to This Study |
|---|---|---|---|---|
| 2 | NASA | 2011 | upgrade in summer 2019: Nylon parts, temperature sensor, wedged window | Temperature (Section 3); Field study (direct sun, Section 5.1) |
| 21 | NASA | 2011 | upgrade in 2016: ARC window; POM-H Delrin parts | Laboratory tests of HCHO emissions (Section 4) |
| 32 | NASA | 2016 | ARC window; POM-H Delrin parts | Field study (direct sun, Section 5.1) |
| 46 | NASA | 2015 | upgrade in 2016 ARC window; POM-H Delrin parts | Laboratory tests of HCHO emissions (Section 4) |
| 108 | ECCC | 2016 | ARC window; POM-H Delrin parts | Field measurements (Section 2) |
| 118 | KNMI | 2016 | ARC window; POM-H Delrin parts | Laboratory tests of HCHO emissions (Section 4) |
| 148 | Virginia Tech | 2018 | temperature sensor (April 2019), wedged window; POM-H Delrin parts | Temperature (Section 3); Laboratory tests of HCHO emissions (Section 4); Field study (MAX-DOAS, Section 5.2) |
| 155 | Boston University | 2019 | temperature sensor; wedged window; POM-H Delrin parts | Temperature (Section 3) |
| 165 | EPA | summer 2019 | Nylon parts; temperature sensor; wedged window | Laboratory tests of HCHO emissions (Section 4) |
| 167 | EPA | summer 2019 | Nylon parts; temperature sensor; wedged window | Laboratory tests of HCHO emissions (Section 4) |
| 168 | EPA | summer 2019 | Nylon parts; temperature sensor; wedged window | Laboratory tests of HCHO emissions (Section 4) |

ECCC: Environment and Climate Change Canada; KNMI: Royal Netherlands Meteorological Institute; EPA: US Environmental Protection Agency; NASA: US National Aeronautics and Space Administration

columns from the Pandora direct sun measurements. Due to ARC degradation, the front window was again replaced in 2018 with a wedged window which practically removed the etaloning interference.

The May-June 2016 Korea–United States Air Quality Study (KORUS-AQ) offered the first opportunity to evaluate direct-sun observations of HCHO total column densities with the improved Pandoras (ARC window and 64 row CCDs, Spinei et al., 2018; Herman et al., 2018). Comparison between the HCHO total columns derived from the Pandora direct sun measurements and the integrated in situ aircraft measurements by Spinei et al., 2018 (Fig. 1) showed that Pandoras overestimated the aircraft derived columns by 16% on average, with an offset of 0.22 DU. However, a point-to-point comparison shows that the measurements agreed on a cold and breezy day (4-May-2016) and on most mornings. Pandoras measured up to 0.8 DU larger columns than DC-8 on hot days during early afternoon hours (12 to 16 hours local time, 1). Measured surface concentrations scaled-up to

**Table 2.** History of Pandora hardware changes related to direct sun HCHO measurements

| Period | Hardware components | Impact on HCHO | HCHO Data Used |
|---|---|---|---|
| 2007 - winter 2016 | parallel window, POM-H Delrin | window caused etalloning in direct sun measurements, HCHO emissions from POM-H Delrin - **direct sun HCHO is not correctable** | MAX-DOAS: Pinardi et al. (2013); Direct Sun: Park et al. (2018) |
| spring 2016 - 2017 | anti reflective coating on parallel widow, POM-H Delrin | ARC degrades within 1 year of installation, temperature dependent HCHO internal emission from POM-H Delrin (disagreement between direct sun total column and MAX-DOAS tropospheric column), can be corrected for functioning ARC | MAX-DOAS: Kreher et al. (2020); Direct Sun: Spinei et al. (2018); Herman et al. (2018); Spinei et al. (2020) |
| 2018 - spring 2019 | wedged window*, POM-H Delrin | temperature dependent HCHO internal emission from POM-H Delrin (disagreement between direct sun total column and MAX-DOAS tropospheric column), can be corrected | MAX-DOAS: Nowak et al. (2020) |
| summer 2019- | wedged widow, nylon | believed not to have any interference caused by design (confirmed by extensive laboratory studies) | |

Note: HCHO from direct sun is not a standard PGN product and was not provided by the NASA and Luftblick PGN groups outside of KORUS-AQ study (Spinei et al. (2018); Herman et al. (2018)). Park et al. (2018) performed HCHO analysis independently and were not aware of any PGN discoveries.

\* wedged windows are installed on new instruments, if the instruments are not returned to NASA or SciGlob - they are not upgraded, therefore some instruments probably still have degrading ARC windows

the total columns, assuming different profiles (black, grey and green lines on 1), and mixing layer height from Ceilometer data agreed with DC-8 measurements better than with Pandora measurements.

Pandora direct sun HCHO total columns were also larger than the multi-axis measured columns during short-term field campaigns (e.g., CINDI-2, LISTOS 2018) and during summer versus winter comparisons for long-term routine observations. Since HCHO is mostly located in the lower troposphere, multi-axis and direct sun measurements should result in HCHO columns that closely match (assuming sampling of the same air masses). DOAS implementation of multi-axis retrieval is significantly less sensitive to instrumental changes. This is due to the fact that single scan sky scattered solar spectra are analyzed using a zenith reference spectrum taken within maximum 10-15 minutes from the scan measurements. Direct sun spectra, on the other hand, are analyzed using a single reference spectrum that was potentially taken months apart from the rest of the spectra. Figure 2 shows an example of HCHO columns derived from Pandora 108 direct sun and multi-axis measurements in Egbert, Canada (44.23°N, -79.78°W) from May 2018 to March 2019. Significant differences (up to 1.5 DU) were observed in retrievals by direct sun measurements in summer during hot days versus the multi-axis measurements.

Comparisons of multiple Pandora data sets covering a variety of ambient conditions led us to conclude that there was an intrinsic property of Pandora that interfered with its HCHO measurements at higher ambient temperatures. The most likely

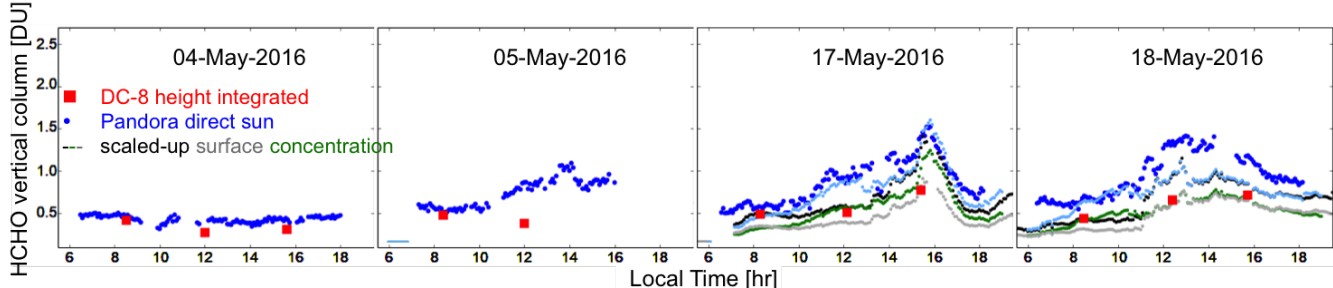

**Figure 1.** HCHO vertical columns during KORUS-AQ near Olympic Park, South Korea derived from direct sun Pandora measurements, DC-8 in situ measurements integrated from surface to 8 km and surface measurements scaled up to tropopause assuming various profile shapes (green: box with a median mixing layer height (MLH), grey: box with a measured MLH; light blue: box+exponential profile with a median MLH, and black: box+exponential profile with a measured MLH, modified from Spinei et al., 2018).

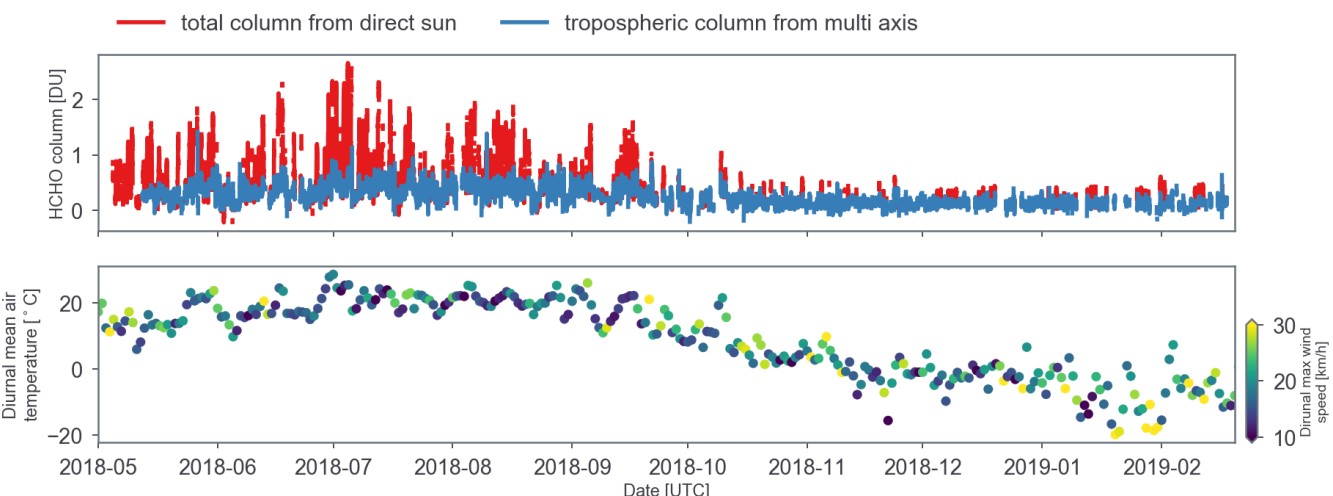

**Figure 2.** Top panel: HCHO direct sun total columns (red) and multi axis tropospheric columns (blue) as retrieved from Pandora 108 near Toronto, Canada, for a 10 months time series 2018/2019. The significant overestimation of direct sun HCHO in summer months is evident. Reference spectrum for direct sun DOAS analysis was collected during a cold winter day. Multi-axis analysis was done with zenith reference spectra measured within 2-3 min of the rest of the spectra. Bottom panel: mean diurnal temperature and maximum wind speed measured near Pandora 108.

source of the observed interference was the Pandora telescope assembly (further referred to as a head sensor). This is the only part of the instrument that was consistently exposed to the ambient conditions without any temperature control.

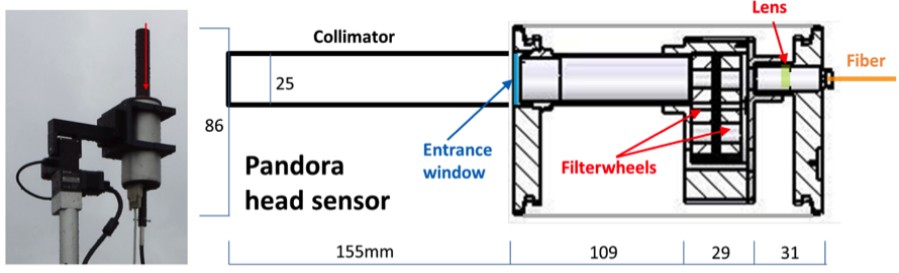

**Figure 3.** Pandora head sensor design (electronics boards are not shown).

## 2.1 Pandora head sensor

The main purpose of the Pandora head sensor is to collect light within a specific field of view, transmit light through optical filters (e.g. U340 to block visible part of solar spectrum), and focus it onto the fiber optics patch cable for transmission to the spectrometer. The Pandora head sensor consists of several components: sealed aluminum cylindrical housing, wedged fused silica entrance window (25 mm in diameter), two filter wheels with motors, baffle holding tubes, lens, fiber optics cable, electronics board (Fig. 3) and a desiccator bag. The baffle holding tube, the two filter wheels, and the dark filter wheel plug were machined from POM-H Delrin, a trade name for polyoxymethylene, engineering thermoplastic up to March of 2019. The desiccant bags (McMaster-CARR model 2189K76, manufacturer Multisorb Technologies, model name MINIPAX) contain activated carbon (43 - 48% by weight) and silica gel (43 - 48% by weight) enclosed in Tyvec material (high-density polyethylene fiber, 5 - 15% by weight) and are designed to remove moisture as well as some VOCs including HCHO.

## 2.2 Polyoxymethylene (POM) Pandora head sensor parts

Polyoxymethylene (POM) has a wide range of applications due to (1) excellent mechanical (high tensile strength, rigidity and toughness) and electrical properties at temperatures from -40 to 130°C (short-term); (2) low reactivity with and low permeability to many chemicals including organic solvents, fuels and gases at room temperature; and (3) ease of processing on standard thermoplastics equipment (Luftl et al., 2014). Six major manufacturers produce about 70% of POM worldwide, and each has its trade name (e.g., Ticona GmbH, Germany:, Celcon®; Polyplastics Co., Ltd., Japan: POM-C Duracon®, Tepcon®; E.I. Du Pont de Nemour & Co., USA: POM-H Delrin®).

POM-H Delrin used for Pandora head sensor parts is a homopolymer POM (POM-H, Ensinger Hyde: black Delrin ecetal resin II150ebk602sheet ¾ x 6 x 6) purchased from McMaster Carr, part numbers: 8575K145 and 8576K21. POM-H is produced by polymerization of purified gaseous formaldehyde in an organic liquid reaction medium in the presence of polymerization catalysts. The resulting polymer has a crystalline granular structure with macromolecules ending in at least one unstable hydroxyl group. These hydroxyl groups are responsible for POM-H thermal instability. POM deterioration occurs mainly due to the following processes:

1. Depolymerization (unzipping);

2. Auto-oxidative scission;

3. Degradation by secondary products of the auto-oxidative scission;

4. Hydrolysis and acidolysis;

5. Photo-oxidation at wavelength 200-800 nm

6. Thermal degradation.

HCHO is a byproduct of most POM degradation processes. Considering the function of the head sensor, we suspect that more than one degradation process will impact POM-H Delrin Pandora head sensor components over their lifetime (several years). POM deterioration studies are typically performed at elevated temperatures (> 90 °C) and focus on mass loss and physical

and mechanical property degradation measurements (Grajales et al., 2015). Review of such literature during the initial Pandora design stage led to the assumption by the NASA and SciGlob teams that HCHO was not emitted from POM-H Delrin under ambient conditions.

It may be worth noting that paraformaldehyde and high-purity $\alpha$-polyoxymethylene have been used to generate known concentrations of HCHO in gaseous mixtures for various applications (Ho, 1985). They are commonly used in permeation

tubes and other permeation devices at elevated temperatures (50-80°C) as a stable source of HCHO for instrument calibrations (Gilpin et al., 1997; Ho, 1985). At elevated temperatures (> 50°C) paraformaldehyde or $\alpha$-polyoxymethylene thermally depolymerizes to produce HCHO vapor that diffuses through the permeation tube membrane.

## 2.3    Pandora internal head sensor temperature

We hypothesized that the thermal instability of POM-H Delrin resulted in HCHO release at higher temperature and was the

source of the temperature-dependent formaldehyde interference in Pandora direct sun HCHO. To test this hypothesis, we added an internal temperature sensor in April 2019 to monitor the internal head sensor temperature in a few instruments. We have evaluated the range of internal head sensor temperatures measured at various sites: Pandora 2 (Greenbelt, MD), 148 (Blacksburg, VA; Rotterdam and Cabauw, the Netherlands) and 155 (Boston, MA) (with an emphasis on the USA East coast where several intergovernmental field campaigns took place, Table 1). Figure 4 shows that internal head sensor temperatures

ranged between 20-25°C during the nighttime hours and up to 45-50 °C during the daytime hours in summer months. During colder months the temperature ranged between 0 to 25 °C. These data suggested that HCHO generation was potentially relevant at internal to head sensor temperatures between 20 and 50 °C.

Internal to head sensor temperatures are determined by the following heat transfer processes between the head sensor and the surrounding environment: (a) Convective heat transfer (natural and forced) due to wind; (b) Radiant heat transfer due to

short wave absorption, long wave emission and long wave absorption; (c) Conduction between the head sensor and the tracker brackets; (d) Energy generation inside the head sensor.

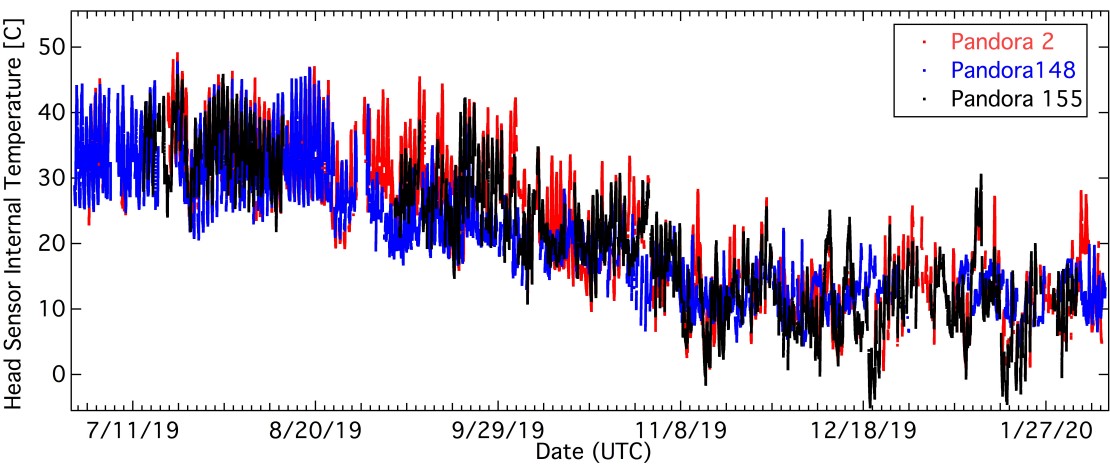

**Figure 4.** Internal head sensor temperatures for Pandora 2 (Greenbelt, MD), 148 (Blacksburg, VA; Rotterdam and Cabauw, the Netherlands) and 155 (Boston, MA)

## 3 Laboratory studies: HCHO columns as a function of Pandora head sensor internal temperature

Since multiple processes are potentially responsible for POM-H Delrin deterioration leading to HCHO generation (see Sec. 2.2) over the instrument lifetime, four Pandora head sensors of various ages were evaluated: 21 (made in 2011), 46 (made in 2015), 118 (made in 2016), and 148 (made in 2018). Pandora 148 was equipped with the internal temperature sensor in April 2019. The other three Pandora head sensors had no internal temperature measurements.

To evaluate the HCHO columns inside the Pandora head sensor as a function of internal temperature, the Pandora head sensor was placed inside a temperature-controlled enclosure ($\pm 0.1$ °C) with a window. Enclosure temperature was varied from 10°C up to 45-55°C, kept at 45-55°C for 0.5-1 hours, and back to 10 °C at different rates. Since POM-H Delrin thermal degradation at higher temperatures is heating rate dependent (e.g., Pielichowska (2015)) we tested atmospheric relevant heating and cooling rates: 3, 3.5, 5, 8, 8.2 °C/hr. Pandora 148 also was tested at 50 °C for 9 hr to determine time needed to reach steady state conditions.

The DOAS technique was used to analyse laboratory measurements to determine differential HCHO columns at various temperatures with respect to the lowest temperature. The experiments were mimicking temperature ranges and DOAS analysis during a typical summer field campaign but without the actual atmosphere. Since the laboratory measurements were performed under tightly controlled conditions (e.g., stable smooth FEL source, constant spectrometer temperature, single gas), the DOAS equation is simplified to Eq. (1) for such conditions:

$$\ln\left(\frac{I_o(\lambda, \mathrm{T}_o)}{I(\lambda, \mathrm{T}) - \mathrm{offset}(\lambda)}\right) = \sigma(\lambda, 298\mathrm{K})_{\mathrm{HCHO}} \cdot \Delta S_{\mathrm{HCHO}}^{\mathrm{T}} + P_L \tag{1}$$

Where, $I(\lambda)$ are the measured attenuated FEL intensities (corrected for instrumental properties) within the wavelength fitting window between 332 to 360 nm at an internal head sensor temperature T; $I_o(\lambda)$ are the measured FEL intensities at internal head sensor temperature $T_o$ corresponding to ambient temperature 10 °C; $\sigma(\lambda, 298K)$ is HCHO molecular absorption cross-section at temperature 298K (Meller and Moortgat, 2000); polynomial order $P_L$ = 5; and offset order 1. This approach estimates differential slant column densities of HCHO along the head sensor length (153.5mm) from the front window to the lens ($\Delta S_{HCHO}^{T} = S_{HCHO}^{T} - S_{HCHO}^{T_o}$).

## 3.1 Experimental setup

The head sensor collimator was protruded through the enclosure window to avoid measuring any potential HCHO outgassing inside the enclosure itself (e.g., paint). The laboratory hosting the measurements was temperature-controlled (20-23°C), has both an air supply and air intake, and the door to the room was open to improve ventilation.

The collimator was pointed at the Gooch and Housego 1000W FEL lamp controlled by the current precision source (OL 410-1000). The FEL lamp was operated at 8A. The distance between the FEL and the collimator was 50 cm.

The enclosure temperature was controlled by NesLab 7 recirculating bath and a LYTRON heat exchanger with two fans to ±0.1°C. The NesLeb 7 temperature sensor was placed near the Pandora head sensor. Enclosure air temperature near the head sensor, front plate Pandora head sensor temperature, and Pandora head side temperature were recorded during all the measurements using fast response stick-on surface thin film PT100 RTD elements (3-Wire, The Sensor Connection). An ADAM-4015 6-channel RTD Module with Modbus digitized the RTD signal. The temperature inside the newer generation head sensor (since April 2019) was measured using the pre-installed Bosch BME280 digital humidity, pressure and temperature sensor on Spark-Fun Atmospheric Sensor Breakout Board. PT100 RTD elements were inter calibrated. They agreed within the manufacture specifications (< 0.15 °C). The accuracy of BME280 was harder to verify due to Pandora head internal power generation of 2W (manufacturer reported accuracy is ±1.0 °C between 0 and 65 °C).

Since only one of the tested Pandora head sensors was equipped with the internal temperature sensor, we determined an outside measurement that is the most representative of the internal temperature. This was done by comparing surface temperature measurements by the PT100 RTD elements at various locations on the Pandora head sensor versus the internal to head sensor temperature. As expected, there is some time delay in response between the surface measurements and the internal temperature. This delay is rate specific. Strong linear correlation between the surface measurements and internal temperature measurements (accounting for transient heat transfer) was observed for both front plate (slope = 0.970, intercept = 4.74°C, RMSE = 0.059 °C) and side (slope = 0.999, intercept = 6.79°C, RMSE = 0.087°C). Since the electronics board heat sink is connected to the front , we use the front plate surface temperature as the proxy for the internal temperature.

The Pandora spectrometer temperature was controlled using a Pandora thermoelectric controller at the set temperature of 15°C. The measurements were averaged over 40 seconds sequentially switching between open, plug, U340, plug filter wheel positions to simulate direct sun measurements.

In addition to the FEL lamp one head sensor (Pandora 118) was also analyzed using a 300 nm LED (Thorlabs M300L4) controlled by a high precision LED driver (Thorlabs DC2200). In the case of the LED source, the Pandora collimator pointed into an 8.3 cm Labsphere Spectralon® reflectance material integrating sphere illuminated by the LED.

## 3.2 Post-summer 2019: "POM-H Delrin free" Pandora head sensors

Since Summer 2019, new Pandora head sensors are POM-H Delrin free. POM-H Delrin was replaced with Molybdenum Disulfide ($MoS_2$) filled Easy-to-Machine Wear-Resistant Cast Nylon 6/6 also purchased from McMaster (Tecamid 66 MO, polyamide > 90% by weight, $MoS_2$ < 10 % by weight, manufactured by ENSINGER INC). To evaluate potential thermal oxidation of polyamide and $MoS_2$ by air oxygen three new head sensors (Pandora 165, 167 and 168, manufactured at the end of 2019) upgraded with the Nylon parts were tested using FEL (1000W) and 300 nm LED sources. The enclosure temperature varied from $10°C$ to $55°C$ over 8 hours, with 1 hour at 55 °, and 8 hours cooling from $55°C$ to $10°C$. This translated to internal temperatures from $17°C$ to $60°C$. The FEL current was set at 7.5A. Pandora spectra were binned within 40s for measurements with no filters (open, single spectrum integration time 2.4 ms, about 12550 cycles per measurement and dark 2320 cycles); 240 s with U340 filter (integration time 12.9 ms, about 15335 cycles per measurement and dark 2835 cycles); and 240 s with BP300 filter (integration time 117 ms, about 1730 cycles per measurement and dark 320 cycles). The spectrometer electronics board temperature was maintained at 12.9 °C (controller set temperature 5 °C). We also repeated the test with 300 nm LED at a constant current of 350 mA with no filters for 100 s total integration time. The experiments were designed to ensure low noise in case of small emissions of HCHO or presence of other species.

The reference spectrum was collected at the lowest internal head sensor temperature (about $17°C$). DOAS fitting included only HCHO and sulfur dioxide, $SO_2$, molecular absorption cross-sections (Vandaele et al., 1998), the polynomial order for the broadband attenuation ($P_L$) was set to 5, and a 0 order polynomial represented the offset. Only the dark current spectrum was subtracted from the data since all other parameters are expected to be constant (e.g., no wavelength shift in the temperature-controlled lab, no need for non-linearity correction, no need for pixel response non-uniformity correction, etc.). The DOAS fitting was performed using QDOAS v3.2 program. $SO_2$ was fitted as a precaution since sulfur containing in $MoS_2$ has been reported to oxidise to $SO_2$ at temperatures higher than 140 °C (https://core.ac.uk/download/pdf/10884897.pdf).

## 4 Laboratory studies results

### 4.1 POM Pandora head sensor HCHO dynamics rates

Analysis of Pandora 148 data over 9-hr period showed that the equilibrium between HCHO generation and removal processes inside the head sensor is reached almost instantaneously (at the DOAS fitting accuracy). Investigation of the actual process mechanisms are outside of the scope of this paper. However, it is probably also controlled by desiccant activated carbon adsorption as a function of temperature, in addition to "pure" solid POM-H - vapor phase processes (Sec. 2.2).

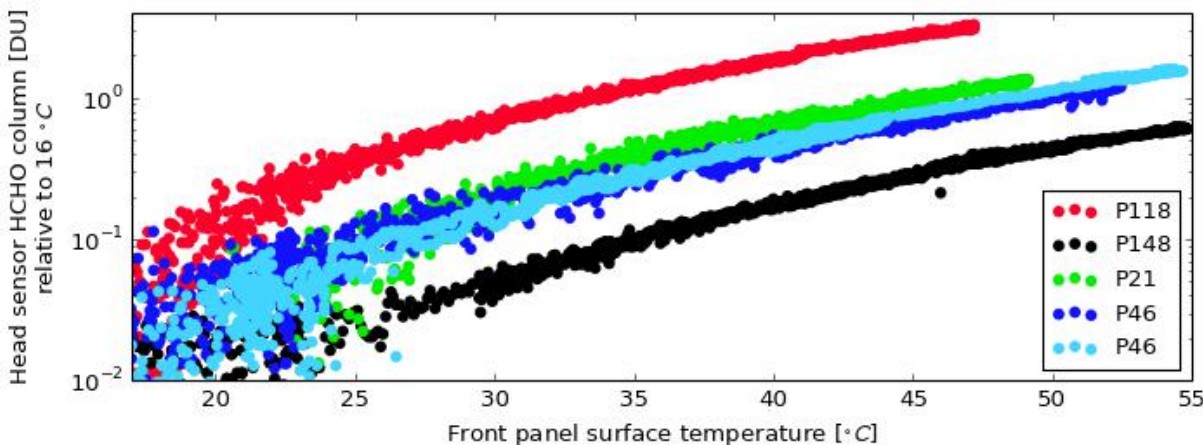

**Figure 5.** HCHO formation/deposition inside Pandora 118, 148, 21 and 46 as a function of front panel outside surface temperature relative to the individual head sensor measurements at 16 °C. Pandora 46 was tested twice with a combination of temperature change rates (3 and 8.2 °C · hr$^{-1}$ light blue and 3.5 °C · hr$^{-1}$ dark blue)

.

Figure 5 shows that HCHO columns inside Pandora head sensors follow exponential dependence on temperature irrelevant of heating or cooling rates for all four tested head sensors (Eq. 2). This temperature dependence does not show any hysteresis at the time scales relevant to this study. HCHO columns in all four sensors had the same exponential function damping ($b = 0.0911 \pm 0.0024$ °C$^{-1}$) but different amplitudes (and most likely absolute offsets).

$$\Delta S(\Delta \mathrm{T_{hs}}) = a \left[ \exp(b \cdot \mathrm{T_{hs}}) - \exp(b \cdot \mathrm{T_{hs}^{ref}}) \right] \tag{2}$$

The newest head sensor (Pandora 148) produced the lowest amount of HCHO (about 0.3 DU) at an internal temperature of about 50 °C (front panel external temperature 45 °C). Pandora 118 generated the largest - about 3 DU at the same temperature. Pandora 46 and 21 were in between. No clear trend was observed between the age of the instruments and the HCHO amount produced in the head sensor. Pandora 148 head sensor was evaluated for temperature dependence of HCHO several times over
five months and did not show any difference in the HCHO generation during that period.

### 4.2 POM-H Delrin free Pandora head sensors - no HCHO production

Initially we conducted DOAS fitting of HCHO and SO$_2$ absorption within their standard fitting windows 332 - 359 nm and 307 - 328 nm (Spinei et al., 2010) respectively. No HCHO or SO$_2$ was detected above the optical depth rms noise level of $5 \cdot 10^{-5}$ from the new "POM-H Delrin free" head sensors as a function of internal head sensor temperature relative to 17°C.
To consolidate HCHO/SO$_2$ results for both species and to evaluate residuals we have done DOAS fitting at a broader fitting window: 300 - 350 nm. Figure 6 shows a retrieval example for Pandora 167 (300 - 350 nm fitting window from spectra collected

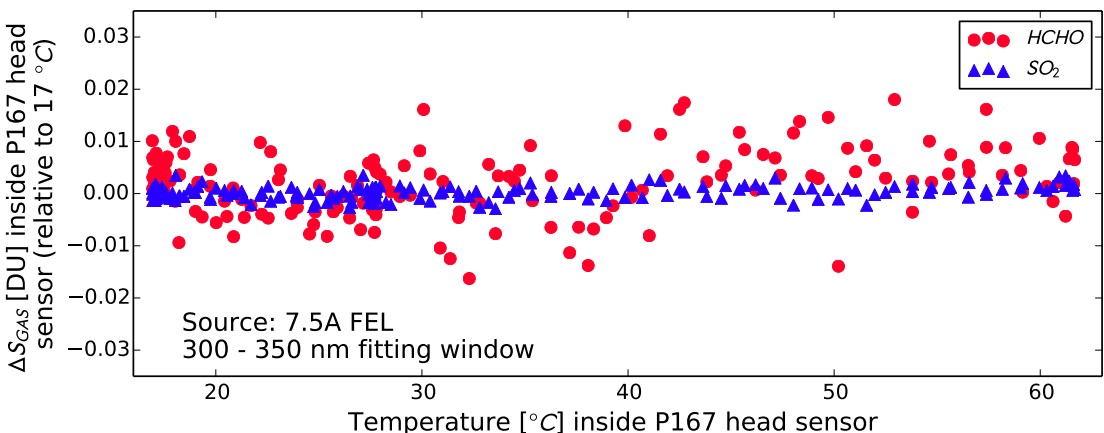

**Figure 6.** $\Delta S$ of HCHO and $SO_2$ retrieved from POM-H Delrin free Pandora 167 head sensor at internal temperatures from $17°C$ to $61°C$ using FEL at 7.5 A (see text for information about thermal rates). Individual spectra integration time 12.9 ms, total exposure per measurement 240 s. Fitting window 300 to 350 nm. Fitting residual optical depth rms was $4.77^{-5} \pm 4.29^{-6}$ during the entire measurements period.

with U340 filter). The spectra were also evaluated for any absorption across the entire instrument wavelength range from 300 to 530 nm by only taking the radiance ratio but not fitting any trace gases. We did not see any signatures above the instrumental noise level.

## 5   Effect of internally generated HCHO on direct sun and multi-axis Pandora HCHO measurements.

In general, DOAS analysis will cancel any instrumental "artifacts" if they are the same in the reference spectrum and the rest of the spectra. As applied to the internally generated HCHO, it will cancel if the reference spectrum and the rest of the spectra are measured at the same head sensor temperature. DOAS fitting results from direct sun or multi-axis measurements ($\Delta S(\mu, \Delta T_{hs}, t)_{HCHO}$) accounting for internal to the head sensor HCHO ($S(T_{hs}, t)^{hs}_{HCHO}$) can be described by the following equation:

$$\begin{aligned}
\Delta S(\mu, \Delta T_{hs}, t) &= S(T_{hs}, t)^{hs} + S(\mu, t)^{atm} - S(T_{hs}^{ref}, t^{ref})^{hs} - S(\mu^{ref}, t^{ref})^{atm} \\
&= S(\mu, t)^{atm} - S(\mu^{ref}, t^{ref})^{atm} + a \cdot \exp(b \cdot T_{hs}) - a \cdot \exp(b \cdot T_{hs}^{ref}) \\
&= \Delta S(\mu, t)^{atm} + a \left[ \exp(b \cdot T_{hs}(t)) - \exp(b \cdot T_{hs}^{ref}) \right]
\end{aligned} \tag{3}$$

Where, $a$ is a head sensor dependent amplitude and is not known for instruments not tested in the laboratory; $b$ is constant damping for all tested systems and is $\approx 0.10 \, °C^{-1}$.

For multi-axis observations where spectra measured at low elevation angles ($\mu$) are analyzed using a zenith ($\mu^{ref}$) reference spectrum measured within a few minutes when $T_{hs} \approx T_{hs}^{ref}$, HCHO amount due to POM-H Delrine emission is about the same and mostly cancels (see Sec. 5.2).

During the CINDI-2 campaign, however, the data analysis protocol for $\Delta S_{\text{HCHO}}$, that were inter-compared between the instruments, was to use a reference spectrum collected around the local noon for all spectra measured throughout the entire day (Kreher et al., 2020). In this case $T_{\text{hs}} \neq T_{\text{hs}}^{\text{ref}}$ and the retrieved $\Delta S_{\text{HCHO}}$ are impacted by the internally generated HCHO
(see Sec. 5.3).

In the case of direct sun measurements, a single reference spectrum at a specific temperature is applied to analyse the data over extended periods. In this case $T_{\text{hs}}(t) \neq T_{\text{hs}}^{\text{ref}}$ and total vertical column ($C$) derived from direct sun measurements is impacted differently depending on the actual head sensor temperature and air mass factor ($AMF$), according to Eq. (4). It is assumed that calibration approach called 'Minimum Langley Extrapolation Method' (Herman et al., 2009) is capable to
estimate the amount in the reference spectrum (including the head sensor amount).

$$
\begin{aligned}
C &= \frac{\Delta S(\mu, \Delta T_{\text{hs}}, t) + S(\mu^{\text{ref}}, t^{\text{ref}})}{AMF(\mu)} = \frac{\Delta S(\mu, t)^{\text{atm}} + \Delta S(T_{\text{hs}}, t)^{\text{hs}} + S(T_{\text{hs}}^{\text{ref}}, t^{\text{ref}})^{\text{hs}} + S(\mu^{\text{ref}}, t^{\text{ref}})^{\text{atm}}}{AMF(\mu)} \\
&\approx \frac{\Delta S(\mu, t)^{\text{atm}} + S(\mu^{\text{ref}}, t^{\text{ref}})^{\text{atm}}}{AMF(\mu)} + \frac{a \cdot \exp(b \cdot T_{\text{hs}}(t))}{AMF(\mu)}
\end{aligned} \tag{4}
$$

### 5.1 Case study: direct sun measurements using collocated POM- and POM-H Delrin free instruments.

Since direct measurements of HCHO in the head sensor is harder to perform during routine direct sun observations, we evaluate the head sensor HCHO production effect using two instruments: Pandora 32 and Pandora 2 (neither was evaluated in
the laboratory according to Sec. 3) during outdoor operation. Both instruments operated side-by-side at the NASA Goddard Space Flight Center in Greenbelt, MD (38.9926°, -76.8396°, 90 m a.s.l.) in direct sun mode during July 2019 - January 2020. Pandora 2, originally built in 2009 with the standard POM-H Delrin components, was upgraded in June 2019 with POM-H Delrin free components and internal temperature sensor (see section 3.2). The Pandora 32 head sensor, originally built in 2012, still contains the original POM-H Delrin components and does not have an internal temperature sensor. To evaluate the effect
of internally generated HCHO on the direct sun total column measurements during a "typical" field campaign study, we used 1.5 month of data from August 30 to October 15, 2019 when both instruments had minimal instrumental issues.
The evaluation consists of several steps:

1. Use Pandora 32 and 2 data to estimate the exponential HCHO production amplitude inside Pandora 32 during selected 1.5 months;

$$
\quad a = \text{median}\left[\frac{\Delta S(\mu, \Delta T_{\text{hs}}, t) - \Delta S(\mu, t)^{\text{atm}}}{\exp(b \cdot T_{\text{hs}}(t)) - \exp(b \cdot T_{\text{hs}}^{\text{ref}})}\right] = \text{median}\left[\frac{\Delta S(\mu, \Delta T_{\text{hs}}, t)^{\text{P32}} - \Delta S(\mu, t)^{\text{P2}}}{\exp(b \cdot T_{\text{hs}}(t)) - \exp(b \cdot T_{\text{hs}}^{\text{ref}})}\right] \tag{5}
$$

2. Calculate HCHO column produced in the head sensor knowing Pandora 2 head sensor temperature and exponential damping and amplitude for Pandora 32 head sensor ($S_{\text{HCHO}} = a \cdot \exp(b \cdot T_{\text{hs}})$) for 7 months (17 July 2019 - 7 Feb 2020);

3. Apply air mass factor to the amount in the head sensor to evaluate diurnal and seasonal contribution to the total column measurements from direct sun data during 7 months (17 July 2019 - 7 Feb 2020).

The assumption about the same internal temperature for Pandora 32 and 2 is based on almost identical head sensor designs, collocation and the same mode of operation. The derived HCHO production amplitude for Pandora 32 is 0.0133 DU. DOAS analysis to calculate HCHO columns was performed in the fitting window 332 -359 nm with $P_L = 4$ and an offset and wavelength shift of polynomial order 1. In addition to HCHO at 298 K (Meller and Moortgat, 2000), absorption by ozone ($O_3$, at 223 and 243 K, Malicet et al., 1995), nitrogen dioxide ($NO_2$, linear temperature model, Vandaele et al., 1998), oxygen collision complex ($O_2O_2$, at 294 K Thalmann and Volkamer 2013), and bromine monoxide (BrO, at 223 K Fleischmann et al., 2004) was fitted. Their high-resolution molecular absorption cross-sections were convolved with the Pandora instrument transfer function prior to DOAS fitting (for convolution details see Cede (2019)). The reference spectrum was created by averaging all spectra within $\pm 5°$ of the minimum SZA on a cloud-free day 15 October 2020 with an average internal head sensor temperature of $\approx 29 \, °C$.

Figure 7 shows a linear correlation between $\Delta S_{\mathrm{HCHO}}$ measured by Pandora 32 and differential columns estimated from Pandora 2 measurements and HCHO produced by the Pandora 32 head sensor. The linear regression analysis between these data sets shows that the exponential function represents a reasonable estimation of the internally generated HCHO by Pandora 32 head sensor measurement during direct sun measurements (slope = 1.00, intercept = -0.03DU and $R^2 = 0.92$ ). Deviations between the true Pandora 32 measurements and simulated from Pandora 2 measurements and internally produced HCHO are also due to small differences in Pandora 32 and 2 fields of view, diffusers, and pointing accuracy.

Figure 8 shows estimated HCHO column density inside Pandora 32 head sensor (red) based on the exponential function coefficients (a = 0.0133 DU, b = 0.0911 $°C^{-1}$) and collocated Pandora 2 internal temperature. Internally generated HCHO amount is smaller during the winter months (< 0.2 DU) and reaches up to 1.15 DU during hot summer days for Pandora 32 head sensor. Since the total $S_{\mathrm{HCHO}}$ is divided by the direct sun air mass factor, the head sensor contribution to the total vertical column is also solar zenith angle-dependent in addition to the internal head sensor temperature (this should not be confused with the actual amount in the head sensor). Its contribution is the largest during the middle of the day near the summer solstice and smallest at large solar zenith angles (80 ° in this study). Due to colder temperatures and larger AMFs during winter months over non-tropical regions, head sensor generated HCHO contribution to the vertical column is small (< 0.1 DU for this instrument). Figure 9 shows that during the cooler and windy summer days (e.g., 23 August 2019), head sensor HCHO can result in a relatively small amount contributing to the total direct sun column. Similar behavior was observed during the KORUS-AQ field campaign (Spinei et al., 2018) when altitude integrated in situ aircraft measurements mostly agreed with Pandora columns in the morning at higher solar zenith angles and disagreed during the middle of the day. Figure 10 (a) shows that warm season measurements at solar zenith angle < 60 ° are most impacted. Figure 10 (b) shows that a significant number of measurements have a contribution to the order of background level ($\approx 0.5$ DU) or higher, which significantly impacts the accuracy of direct sun observations.

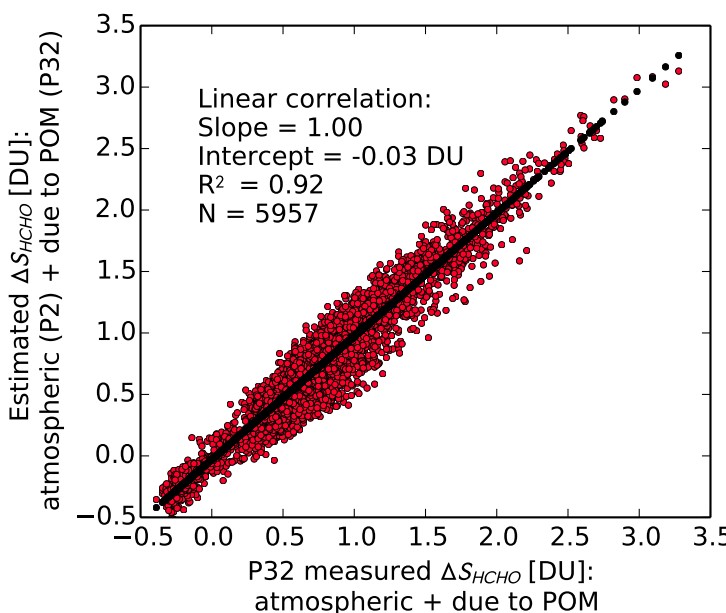

**Figure 7.** Linear regression analysis of the estimated and measured $\Delta S_{\mathrm{HCHO}}$ by Pandora 32, including true atmospheric and POM-H Delrine emitted HCHO. Pandora 2 (POM free) measured only atmospheric HCHO. Pandora 32 measured both true atmospheric abundance and POM-H Delrine emitted HCHO. Instruments were collocated at NASA/GSFC and made direct sun measurements from 30 August to 15 October 2019. The reference spectrum was collected around local noon on 2019/10/15: $\Delta S_{\mathrm{HCHO}}^{\mathrm{hs}} = 0.0133 \cdot \exp(0.0911 \cdot \mathrm{T_{hs}})$ DU

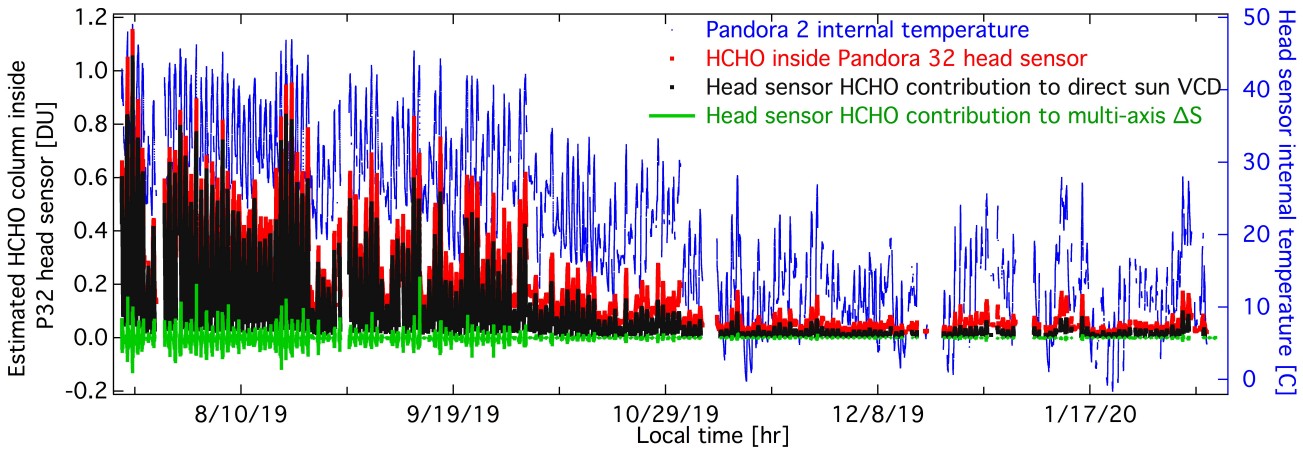

**Figure 8.** Estimated HCHO column density inside Pandora 32 head sensor during deployment at GSFC/NASA, Greenbelt, MD. Estimation is based on the exponential function amplitude derived from Pandora 32 and Pandora 2 direct sun measurements of HCHO (0.0133 DU) and exponential function damping coefficient derived from the laboratory measurements of four other instruments (0.091). Direct sun air mass factors used to calculate HCHO are limited to solar zenith angles smaller than $80°$

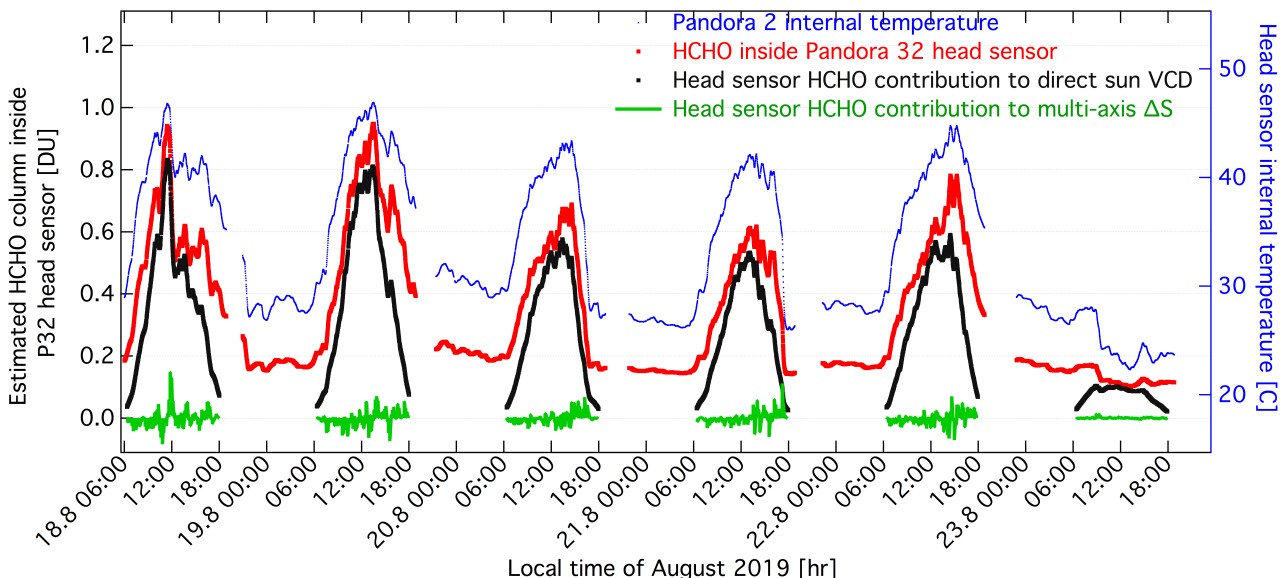

**Figure 9.** Estimated HCHO column density inside Pandora 32 head sensor during deployment at GSFC/NASA, Greenbelt, MD. Estimation is based on the exponential function amplitude derived from Pandora 32 and Pandora 2 direct sun measurements of HCHO (0.0133 DU) and exponential function damping coefficient derived from the laboratory measurements of four other instruments (0.091). Direct sun air mass factors used to calculate HCHO are limited to solar zenith angles smaller than 80°

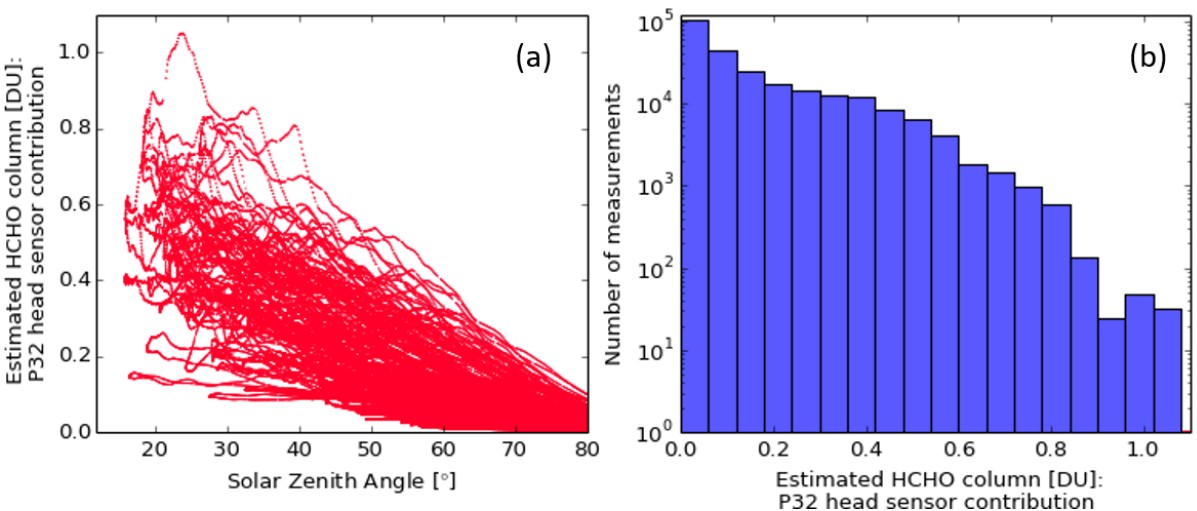

**Figure 10.** Estimated HCHO column density inside Pandora 32 head sensor during deployment at GSFC/NASA, Greenbelt, MD (17 July 2019 - 7 Feb 2020). Estimation is based on the exponential function amplitude derived from Pandora 32 and Pandora 2 direct sun measurements of HCHO (0.0133 DU) and exponential function damping coefficient derived from the laboratory measurements of four other instruments (0.091). Direct sun air mass factors used to calculate HCHO are limited to solar zenith angles smaller than 80°

## 5.2 Case study: effect of internally generated HCHO on multi-axis measurements during TROLIX'19 field campaign

In this section we evaluate the variability of internally generated HCHO, and its effect on MAX-DOAS retrievals using "clos-
est" in time (< 10 minutes) zenith reference spectra during TROLIX'19 campaign. Pandoras 148 and 118 participated in
TROLIX'19 campaign. The main goal of TROLIX'19 was validation of TROPOMI L2 main data products including UVAI,
Aerosol Layer Height, $NO_2, O_3$, and HCHO under a wide range of atmospheric conditions. Pandora 148 has been tested in
the laboratory (see Sec. 4.1) three times over the period of 5 months and showed no changes in internally produced HCHO as
a function of temperature. We use Pandora 148 data collected during the TROLIX'19 campaign to estimate the effect of inter-
nally produced HCHO on the multi-axis $\Delta S_{HCHO}$ retrieved with individual scan reference. Pandora 148 was equipped with an
internal temperature sensor and had well characterized internal HCHO temperature dependence before deployment in western
Rotterdam metropolitan area (51.9172°, 4.4066°, 7 m above sea level) during September 2019. Pandora 118 was characterized
for temperature dependent HCHO production in December 2019, three months after the TROLIX'19 deployment.

The effect of internally generated HCHO on multi-axis $\Delta S_{HCHO}$ using closest in time reference zenith spectrum collected
maximum 10 min apart from the rest of the scan spectra are evaluated according to the following steps:

1. Calculating the head sensor produced amount, $S_{HCHO}(T_{hs}(t))$, based on the head sensor temperature and Pandora 148
   exponential function coefficients: a = 0.0041 DU and b = 0.0911 °C$^{-1}$;

2. Calculating the head sensor HCHO produced amount at the individual scan reference spectrum time (maximum 10
   min) based on the head sensor temperature and Pandora 148 exponential function coefficients: a = 0.0041 DU; b =
   0.0911 °C$^{-1}$ ($S_{HCHO}(T_{hs}^{ref})$);

3. Calculate the amount of HCHO due to POM: $\Delta S_{HCHO}^{hs} = a \left[ \exp(b \cdot T_{hs}(t)) - \exp(b \cdot T_{hs}^{ref}) \right]$

Figure 11 shows that Pandora 148 internally generated HCHO contribution to the multi-axis $\Delta S_{HCHO}$ while using single
scan reference is very small (< 0.005 DU). As expected, this small contribution is mostly due to lower internal temperature
variations within 10 min period and partially due to small generation rates inside Pandora 148 head sensor (a = 0.0041 DU,
see Fig. 5). Since Pandora 118 was only characterized once for temperature dependence of HCHO production we do not have
high confidence in its temperature dependence "stability". If we assume that the exponential function amplitude was the same
in September as in December (a = 0.049 DU), Pandora 118 head sensor contributed almost 10 times more than Pandora 148 to
multi-axis $\Delta S_{HCHO}$. Even in this case, the resulting amount is smaller than 0.05 DU, which is lower than the DOAS fitting
noise for most DOAS instruments (0.3 DU = $8 \times 10^{15}$ molecules/cm$^2$, Table 7 in Kreher et al. (2020).)

## 5.3 Case study: effect of Pandora internally generated HCHO on the CINDI-2 $\Delta S_{HCHO}$ intercomparison with other DOAS instruments

Five Pandoras participated in the Second Cabauw Intercomparison campaign for Nitrogen Dioxide measuring Instruments
(CINDI-2) that took place at Cabauw, The Netherlands (51.97° N, 4.93° E, September 2016) (Kreher et al., 2020). A formal

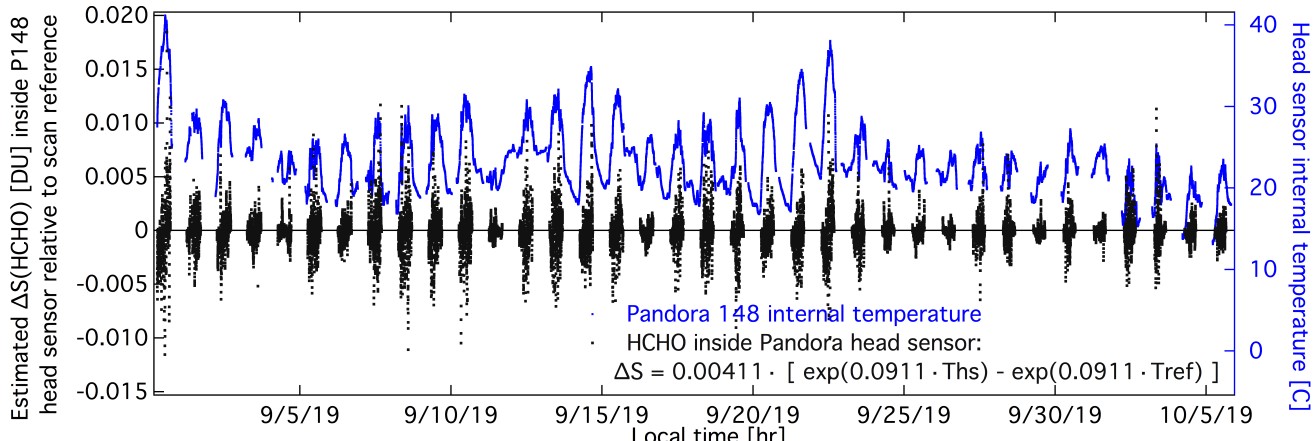

**Figure 11.** Estimated contribution from internally generated HCHO on the multi-axis $\Delta S_{HCHO}$ measured by Pandora 148 during TROLIX'19 using individual scan reference spectrum about 10 min apart from the lowest viewing angle.

semi-blind intercomparison exercise was performed to compare $\Delta S_{gas}$ of $NO_2$, HCHO, $O_2O_2$, and $O_3$ measured by 36
spectroscopic systems from 24 institutes during 17 days in September 2016. To limit any variability due to differences in temporal sampling by each instrument for semi-blind intercomparison exercise, all multi-axis daily scans were analyzed using that day's local noon spectra. This type of analysis results in a stronger contribution of the internally generated HCHO on the Pandora $\Delta S_{HCHO}$ that were compared with the rest of DOAS instruments. Since none of the Pandoras in September 2016 were equipped with an internal temperature sensor, we use Pandora 148 data during TROLIX'19 measurements as a surrogate for
the CINDI-2 campaign. Pandora 148 was deployed at a location about 38 km south-west of the CINDI-2 site during the same month of the year as CINDI-2. While differences in atmospheric conditions are expected between the sites and years (2019 vs. 2016), we assume that general trends in internal Pandora head sensor temperature are similar over the measurement periods. Only one Pandora (Pandora 118) was tested for internal HCHO generation, but more than three years later. We assume that Pandora 32 exponential amplitude is more representative of a "typical" Pandora rate than Pandora 118.

To evaluate the effect of internally generated HCHO on $\Delta S_{HCHO}$ used for semi-blind intercomparison during CINDI-2, a single reference spectrum was used to analyze the entire day of multi-axis data during TROLIX'19 by:

1. Calculating the head sensor produced amount based on Pandora 148 head sensor temperature and Pandora 32 exponential function coefficients: a = 0.013 DU; b = 0.0911 $°C^{-1}$;

2. Calculating the head sensor produced amount at the reference spectrum time (local noon at minimum solar zenith angle);

3. Calculate $\Delta S_{HCHO}^{hs}$ due to POM.

Figure 12 shows that the internally generated HCHO contribution to $\Delta S$ is negative before local noon, positive during early afternoon and negative again during later afternoon. Since the internal temperatures did not vary by more than 15°C during

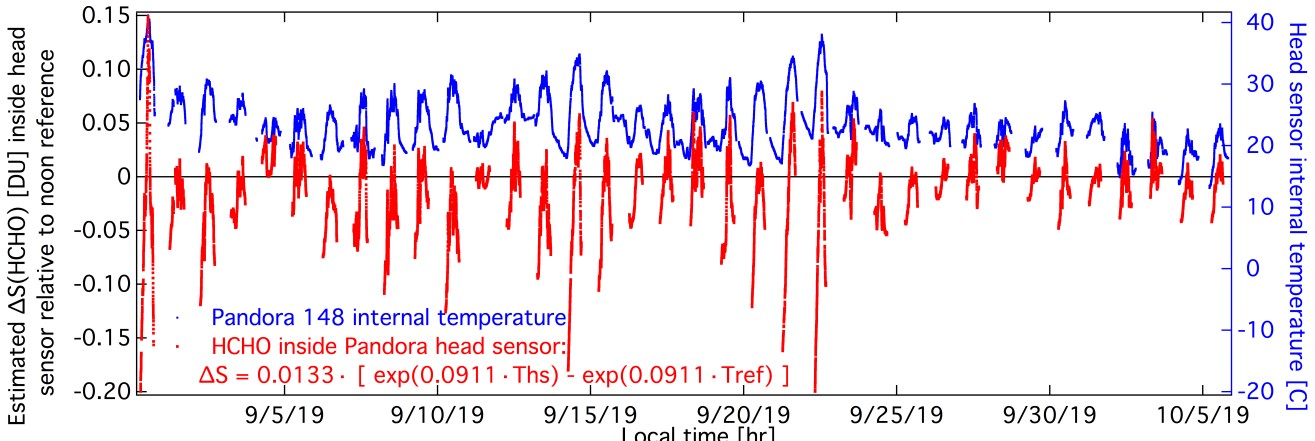

**Figure 12.** Estimated $\Delta S_{\mathrm{HCHO}}$ contributing to the multi-axis measurements while local noon reference is used. This is relevant only to DOAS instrument intercomparison campaigns such as CINDI-2 since standard data inversion requires individual scan reference. Exponential production amplitude applied is 0.0133 DU, however the actual amplitude observed in the lab was between 0.0041 and 0.049. $\Delta S_{\mathrm{HCHO}}$ is calculated based on Pandora 148 head sensor temperature during TROLIX'19 campaign west of Rotterdam, the Netherlands during August 31 to 5 October 2019.

daily measurements and maximum did not exceed $40°$C the overall effect is in general small $< 0.1$ DU with slightly negative bias (Fig. 13) for an instrument similar to Pandora 32, 21 and 46. While we do not know the exact HCHO internal generation rates for the Pandoras deployed during CINDI-2 we can assume that the minimum corresponds to Pandora 148 and maximum to Pandora 118, which is about 3.2 times smaller or 3.7 times larger than in Fig. 12 and 13.

Note, that DOAS analysis using daily noon zenith reference spectra was implemented only for the formal semi-blind inter-comparison of $\Delta SCD$ exercise (Kreher et al., 2020). Full data processing and inversion to the final products, tropospheric columns and profiles, was done using individual scan zenith spectra not daily noon zenith spectra (Tirpitz et al., 2020).

## 6   Conclusions

Pandora direct sun measurements of HCHO were impacted by the internally generated HCHO inside head sensor due to thermal degradation of POM-H Delrin plastic parts up until summer 2019. Direct sun measurements before spring 2016 were also effected by the etaloning off the front window surfaces. Pandora multi-axis measurements of HCHO were significantly less impacted by the internally generated HCHO. The following list represents the major findings of this work:

1. HCHO in Pandora head sensors (up to summer 2019): exponential temperature dependence of HCHO production was observed for four tested head sensors with a damping coefficient of $0.0911 \pm 0.0024 °\mathrm{C}^{-1}$. The exponential function amplitude ranged from 0.0041 DU for P148 to 0.049 DU for P118. No apparent dependency on the head sensor age and heating/cooling rates was observed (Fig. 5);

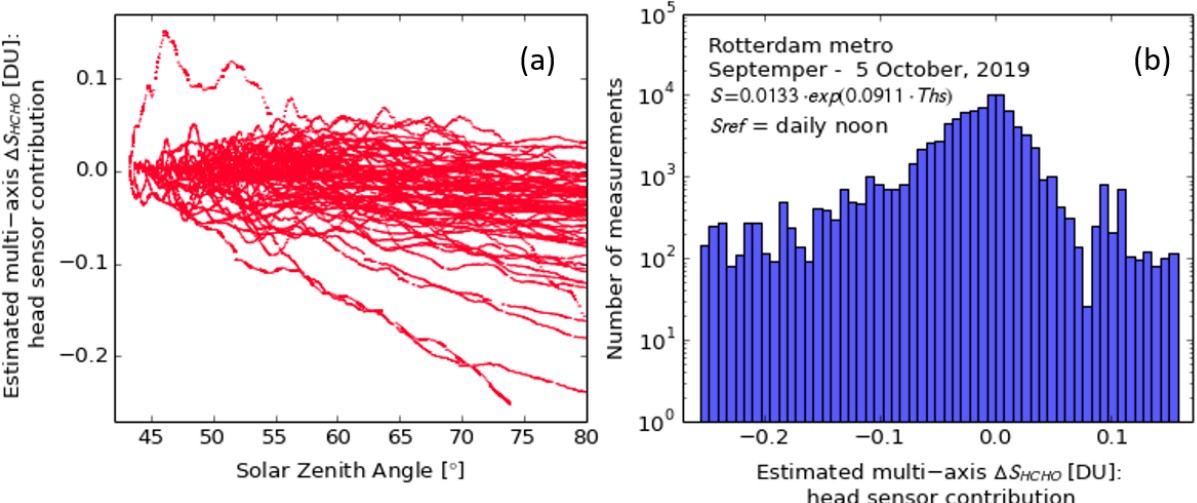

**Figure 13.** Estimated $\Delta S_{\mathrm{HCHO}}$ contributing to the multi-axis measurements while local noon reference is used. This is relevant only to DOAS instrument intercomparison campaigns such as CINDI-2 since standard data inversion requires individual scan reference. Exponential production amplitude applied is 0.0133 DU, however the actual amplitude observed in the lab was between 0.0041 and 0.049. $\Delta S_{\mathrm{HCHO}}$ is calculated based on Pandora 148 head sensor temperature during TROLIX'19 campaign west of Rotterdam, the Netherlands during 31 August to 5 October 2019.

2. Three new POM-H Delrin free Pandora head sensors (starting from summer 2019) were evaluated for temperature depen-
dent attenuation across the entire spectral range. The noise was minimized by reducing spectrometer temperature to 5 °C
set temperature and averaging more than 10000 spectra per measurement to allow detection of smaller absorption (Fig.
6). No HCHO or $SO_2$ were detected under the measurement conditions. No other absorptions above the instrumental
noise were observed across the entire spectral range;

3. Evaluation of $\Delta S_{HCHO}$ measured from two collocated Pandora instruments, one with POM-H Delrin and one with-
out POM-H Delrin parts, operating in direct sun mode allowed for derivation of exponential production amplitude
(0.0133 DU, Fig. 7);

4. The total amount of HCHO internally generated by the POM-H Delrin components and contributing to the direct sun
measurements were estimated based on temperature and solar zenith angle of the measurements. Measurements in winter,
during colder days in general and at high solar zenith angles (> 75 °) were minimally impacted. Measurements during
hot days and small solar zenith angles had up to 1 DU contribution from POM-H Delrin parts (Fig. 8, 9, 10).

5. Pandora HCHO measurements derived from Pandora direct sun observations between 2016 and 2019 cannot be used
in the current form for any scientific conclusions about atmospheric HCHO. Results presented here most likely are
representative of other Pandora instruments operational between 2016 and summer 2019.

Considering that Pandora head sensors have almost identical design from material, shape and thermodynamics point of view measurements between 2016 and 2019 can be corrected based on (a) meteorological observations (temperature and wind) to estimate internal head sensor temperature and (b) on $\Delta S$ measurements to estimate HCHO production amplitude (Spinei et al. 2020 in preparation)

6. Multi-axis measurements had a minimal contribution (< 0.01 DU) to $\Delta S_{HCHO}$ due to the scan reference spectrum and the rest of the scan spectra collected within a short time period with small difference in head sensor temperature (Fig. 11).

7. CINDI-2 instrument intercomparison data analysis (Kreher et al., 2020) is not representative of the final multi-axis data processing (leading to profile inversion described by Tirpitz et al. (2020)) since the noon reference spectrum was used for DOAS fitting and no subsequent subtraction of the scan zenith was done. This resulted in higher internal head sensor contribution to $\Delta S_{HCHO}$ that were inter-compared with the other instruments (Fig. 12, 13).

*Data availability.* All Pandora data are available from lb3.pandonia.net

*Author contributions.* Elena Spinei (ES) has conceived the idea of the laboratory testing, performed laboratory data collection and analysis and taken the lead role in writing the manuscript. ES operated Pandora 148 during TROLIX-19 campaign.

Manuel Gebetsberger and Moritz Mueller assured Pandora network data quality and data analysis from multiple instruments, and edited the manuscript.

Martin Tiefengraber and Alexander Cede lead the pre head sensor HCHO discovery effort to identify HCHO inconsistency issues and edited the manuscript Alexander Kostakis and Fernando Santos contributed to introduction writing, manuscript editing and manuscript logical flow improvement.

Nader Abbuhasan provided custom modifications to the Pandora head sensor, closely worked with ES to ensure Pandora equipment availability.

Luke Valin, James Szykman, and Andrew Whitehill participated in early problem identification and edited the the manuscript

Xiaoyi Zhao, Vitali Fioletov, and Sum Chi Lee operated and managed the Canadian Pandora network, which provided the early field evidence of the problem.

Robert Swap closely worked with ES to ensure Pandora equipment availability for laboratory testing and participated in strategy development.

All co-authors discussed the results.

*Competing interests.* No competing interests are present

*Acknowledgements.* NASA Tropospheric Composition branch for providing Pandora instrumentation for testing. Michel Van Roozendael, BIRA, for a suggestion to look into the plastic components used in the Pandora head sensor. Ankie Piters, KNMI, for providing Pandora 118 for testing. Jeffrey Geddes for contributing Pandora 155 head sensor temperature data. Nabil Nowak, Yun Dong and Nash Kocur, Virginia Tech graduate students, for helping with early testing of Pandora 148. Jonathan Davies and Ihab Abboud, ECCC, for technical supports to Pandora 108.

445

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
