# Peer review of "Effect of Polyoxymethylene (POM-H Delrin) offgassing within Pandora head sensor on direct sun and multi-axis formaldehyde column measurements in 2016 - 2019"

_Atmospheric Measurement Techniques, 2020_

## Referee Comment (RC1) · Anonymous Referee #1 · 1 Oct 2020

The paper presents a Pandora HCHO history and analysis of an internal HCHO creation (up to 1DU, i.e. 2.69 x$10^{16}$ molec/cm$^2$ in worst cases) due to a head-sensor constituent (the POM-H Delrin) present before summer 2019. The study is based on results of several Pandora instruments, deployed in different locations and periods, and on lab measurements. Both direct-sun and MAX-DOAS measurement modes are analysed and discussed. The paper is well written and the topic is within AMT scope and of interest for the community. I recommend publication after some minor additions/revisions below.

General comments

I would recommend adding one or two a tables summarizing 1) which instruments have been used when/for which part of this work, to help the reader following which instruments have been participating to several steps; and 2) the different steps of the Pandora HCHO "history" with some links to published HCHO Pandora papers: - pre-2016: no HCHO possible;

- spring 2016: window replacement and KORUS-AQ and CINDI-2 but issue of coherence between MAX-DOAS and direct-sun results;

- april 2016: added temperature sensors for a few instruments and highlight of the potential link of HCHO creation with temperature;

- lab measurements to confirm the findings;

- summer 2019: new head sensor without POM-H Delrin to solve the problem;

- measurements campaigns illustrating the findings.

In the light of the "pre-2016" first point, there should also be some comments of Pandora HCHO measurements performed before 2016 (e.g., in Pinardi et al. 2013, for measurements in 2009 – is there any other ?). I would also include some more references in the introduction on satellite HCHO (only one is mentioned, while at least 8 are known - and given below) and on satellite validation. In the abstract and/or conclusions, it would also be nice to also convert the findings from DU to molec/cm$^2$, as this unit is usually used for HCHO retrievals.

Specific comments and technical corrections

- P2, line 29-33: give some references for the other satellites, as done for SCIAMACHY. Several are suggested below.

- P2, line 37: give more references for other validation studies. Several are suggested below.

[Figure]

- P2, last line: Lamsal et al. 2014 and Kollonige et al. 2017 are not given in the reference list. Consider adding other recent works with several Pandora NO2 measurements used for validation (Herman et al., 2019; Pinardi et al., 2020; Verhoelst et al., 2020). Please check completeness of the reference list in the whole manuscript (e.g., Reed et al., 2015, Gronoff et al., 2019 are also missing in the references).

- P3, line 65 to 71: it would be nice to link the different steps, with the manuscript sections. Or at least add the link with the sections in the proposed additional table n° 2.

- P3, line 74: Âń Retrieval of weak absorbers such as HCHO was not possible from the pre-2016 Pandora direct sun measurements due to the telescope assembly front window etaloning": comment Pinardi et al., 2013 measurements of 2009.

- P4, figure 1: please revisit the legend. What are the light blue points?

- P4, line 95: . . .the reference spectrum is *often* taken within. . . (not for the CINDI-2, as mentioned later in the paper)

- P5, line 108: "A baffle holding tube, two filter wheels, and a dark filter" –> The baffle holding tube, the two filter wheels, and the dark filter. . .

- P7, line 149: remove parenthesis in "Figure (4). . ."

- P7, line 153: consider changing "follow-up work" to the relevant section of the manuscript.

- P7, line 160: add a comma before 118

- P7, lines 147 to 148 and lines 160 to 161: as proposed in the general comments, consider adding a table with the Pandora numbers, their construction year, and in what part of this work they appear.

- P8, line 161: "Three other Pandora" –> The other three Pandora

- P8, line 169: consider adding here the information on the 1000 W FED lamp given in line 186 and the LED lamp (cf line 2018 and 2016), or removing this information from line 169.

- P8, line 170: what is the meaning/purpose of the "The head sensor was not disturbed during the measurements" ?

- P8, line 175 to 179: these DOAS settings seems the generic ones used in the laboratory study (332-360nm), but actually in P10 line 250, these are different (300-350nm) (and also different than the in-field settings, P13 line 396, 332-359nm), so this is a bit perturbing. Is there any reason why changing wavelength ranges, polynomial and offset order? For the inclusion of trace gases, this is clear/mentioned when relevant.

- P9, line 193: "using the pre-installed Bosch BME280 digital humidity, pressure and temperature sensor on Spark-Fun Atmospheric Sensor Breakout Board" -> this is a detail of the new head sensor. It would be maybe better to introduce it in Sect 2.3 when introducing the Pandora internal head sensor temperature?

- P9, lines 208 to 210: the Pandora 118 was also tested with a LED lamp in addition to the FEL lamp. Is this shown somewhere? Why is this done? What was expected (differently) from this additional test?

- P10 and P11 (figure 6): POM-free and Delrin-free. These are used as synonyms? In P13, line 281, also POM-H is used. Use one name everywhere.

- P12, line 268 and elsewhere in the manuscript: replace Kreher et al., 2019 by Kreher et al., 2020.

- P12, line 269: there are two point at the end of the line.

- P14, line 314: "contribution" seems to be in bold

- P17, line 348: add reference to Fig 5 for the "small generation rates inside Pandora 148 head sensor (a = 0.0041 DU)" ?

- P17, lines 53: give an estimation and a reference (Pinardi et al., 2013 ?) of the "fitting noise for most DOAS instruments" (usually in molec/cm2!).

- P17, line 357: add "Sept. 2016" after the coordinates and update Kreher reference.

- P18, line 378: if I followed well all the Pandora numbers, 32 was involved in the GSFC field campaign, and 31 and 46 in the lab measurements wrt temperature. A table would definitely help follow which instrument has been used when!

- P20: suggestion to add in each bullet conclusions the Pandora numbers relevant to each bullet (or not if it is already clear in an additional table), and the relevant Figure supporting each conclusion (e.g., Fig 5 for the 1rst point). Also add molec/cm$^2$ estimation in addition to DU values.

- P20, point 5: maybe cite publications that were made with Pandora direct sun HCHO data between 2016 and 2019, and that should not be considered "valid"?

Suggested References:

[revised manuscript text omitted]

Wang, Y., Beirle, S., Lampel, J., Koukouli, M., De Smedt, I., Theys, N., Li, A., Wu, D., Xie, P., Liu, C., Van Roozendael, M., Stavrakou, T., Müller, J.-F., and Wagner, T.: Validation of OMI, GOME-2A and GOME-2B tropospheric NO2, SO2 and HCHO products using MAX-DOAS observations from 2011 to 2014 in Wuxi, China: investigation of the effects of priori profiles and aerosols on the satellite products, Atmos. Chem. Phys.,

17, 5007–5033, https://doi.org/10.5194/acp-17-5007-2017, 2017.

Vlemmix, T., Hendrick, F., Pinardi, G., De Smedt, I., Fayt, C., Hermans, C., Piters, A., Wang, P., Levelt, P., and Van Roozendael, M.: MAX-DOAS observations of aerosols, formaldehyde and nitrogen dioxide in the Beijing area: comparison of two profile retrieval approaches, Atmos. Meas. Tech., 8, 941–963, https://doi.org/10.5194/amt-8-941-2015, 2015.
* * *

---

## Referee Comment (RC2) · Anonymous Referee #2 · 8 Oct 2020

The manuscript "Effect of Polyoxymethylene (POM-H Delrin) offgassing within Pandora head sensor on direct sun and multi-axis formaldehyde column measurements in 2016 – 2019" by E. Spinei et al., describes an issues of overestimation of HCHO columns measured by certain Pandora instruments due to thermally-induced HCHO emissions from the material of instruments' sensor heads (manufactured from plastic polyoxymethulene homopolier, POM-H). The paper describes laboratory investigations performed to evaluate the magnitude of the offgassing effect, field co-location case studies, as well as the laboratory and field evolution of the instrument with a new, POM-

free, sensor heads. The manuscript is of interest for atmospheric sciences community, provides a clear definition, and a thorough assessment of the issue. The topic is appropriate for AMT, and I recommend publishing the manuscript after authors address a minor comments outlined below.

General comments:

The manuscript provides a detailed timelines and descriptions for multiple events such as Pandora sensor head design changes; laboratory testing for POM and non-POM units; and case-studies for real-life deployments/colocations. I found myself flipping back and fourth through the manuscript, I therefore think that the manuscript will be improved by inclusion of a master table summarizing the types of tests, dates, intercomparison campaigns, identification numbers of units, modifications to the units, etc. This table can be placed at the end of Section 1. Can authors develop a correction factor that can be applied for direct sun HCHO data collected during 2016-2019 to correct for HCHO production, so the dataset can be utilized by the scientific community? For example, recommending temperature ranges during which data would be usable, and showing examples of intercomparison with in-situ techniques (if available) or with satellite data showing a reasonable agreement.

Specific comments:

Lines 10-14: define cold and warm temperature ranges. Remove quotation marks from "cold"

Figure 1: define light blue, gray and green lines

Lines 91-98: add explanation for which spectra are used for direct sun and multi-axis DOAS retrievals.

Figure 2: add ambient temperature to the figure

Line 160: add a coma before 118

Line 161: add a coma before and 148

Line 171: remove quotation marks from, "mimicking"

Line 314: remove bold face from contribution

Lines 404-405: The statement "Pandora HCHO measurements derived from direct sun observations between 2016 and 2019 cannot be used in the current form. Results presented here most likely are representative of other instruments build between 2016 and 2019" is very drastic. Authors should consider adding recommendations on possible corrective approaches, so the data could be utilized by scientific community.
* * *

---

## Author Comment (AC2) · 31 Oct 2020

We thank Anonymous Referee 2 for the comments and recommendations. Our responses to the referee comments are in italics font.
**0.0.1 General comments:**

The manuscript provides a detailed timelines and descriptions for multiple events such as Pandora sensor head design changes; laboratory testing for POM and non-POM units; and case-studies for real-life deployments/colocations. I found myself flipping back and fourth through the manuscript, I therefore think that the manuscript will be improved by inclusion of a master table summarizing the types of tests, dates, inter-comparison campaigns, identification numbers of units, modifications to the units, etc. This table can be placed at the end of Section 1.

*We have added Table 1 (see the supplement) that summarizes the Pandora instrument information and how they contributed to this study. We also added the following sentence at the end of Section 1: "Table 1 lists the Pandora instruments description and contribution to this study.". We did not add the actual field campaigns that each pandora participated in to make sure we do not give a wrong impression that only these instruments are impacted.*

Can authors develop a correction factor that can be applied for direct sun HCHO data collected during 2016-2019 to correct for HCHO production, so the dataset can be utilized by the scientific community? For example, recommending temperature ranges during which data would be usable, and showing examples of intercomparison with in-situ techniques (if available) or with satellite data showing a reasonable agreement.

*We have performed detailed evaluation of Pandora head sensor heat transfer and estimation of HCHO production amplitude. We are preparing a separate manuscript: Spinei et al. 2020 (in preparation). We added the following reference to point 5 of the conclusions: " Considering that Pandora head sensors have almost identical design from material, shape and thermodynamics point of view*

*data between 2016 and 2019 can be corrected based on (a) meteorological data (temperature and wind) to estimate internal head sensor temperature and (b) on $\Delta S$ measurements to estimate* HCHO *production amplitude (Spinei et al. 2020 in preparation)"*

**0.0.2 Specific comments:**

Lines 10-14: define cold and warm temperature ranges. Remove quotation marks from "cold"

*We added the temperature information: Measurements in winter, during colder ($< 10^oC$) days in general and at high solar zenith angles ($> 75^o$) were minimally impacted. Measurements during hot days ($> 28^oC$)*

Figure 1: define light blue, gray and green lines

*We added the following information to the figure caption: (green: box with a median mixing layer height (MLH), grey: box with a measured MLH; light blue: box+exponential profile with a median MLH, and black: box+exponential profile with a measured MLH,*

Lines 91-98: add explanation for which spectra are used for direct sun and multi-axis DOAS retrievals.

*We have made the following modification: DOAS implementation of multi-axis retrieval is significantly less sensitive to instrumental changes. This is due to the fact that single scan sky scattered solar spectra are analyzed using a zenith*
*reference spectrum taken within maximum 10-15 minutes from the scan mea-
surements. Direct sun spectra, on the other hand, are analyzed using a single
reference spectrum that was potentially taken months apart from the rest of the
spectra.*

Figure 2: add ambient temperature to the figure

*added*

Line 160: add a coma before 118

*added*

Line 161: add a coma before and 148

*added*

Line 171: remove quotation marks from, "mimicking"

*removed*

Line 314: remove bold face from contribution

*removed*

Lines 404-405: The statement "Pandora HCHO measurements derived from direct sun
observations between 2016 and 2019 cannot be used in the current form. Results pre-
sented here most likely are representative of other instruments build between 2016 and
2019" is very drastic. Authors should consider adding recommendations on possible
corrective approaches, so the data could be utilized by scientific community.

*We are preparing a publication to describe such corrections. We added: "Considering that Pandora head sensors have almost identical design from material, shape and thermodynamics point of view measurements between 2016 and 2019 can be corrected based on (a) meteorological observations (temperature and wind) to estimate internal head sensor temperature and (b) on $\Delta S$ measurements to estimate* HCHO *production amplitude (Spinei et al. 2020 in preparation)*

**Supplement:**

**Table 1.** Pandora instruments used in this study

| N | Owner | Manufactured | Relevant Hardware Components | Contribution to This Study |
|---|---|---|---|---|
| 2 | NASA | 2011 | upgrade in summer 2019: Nylon parts, temperature sensor, wedged window | Temperature (Section 3); Field study (direct sun, Section 5.1) |
| 21 | NASA | 2011 | upgrade in 2016: ARC window; POM-H Delrin parts | Laboratory tests of HCHO emissions (Section 4) |
| 32 | NASA | 2016 | ARC window; POM-H Delrin parts | Field study (direct sun, Section 5.1) |
| 46 | NASA | 2015 | upgrade in 2016 ARC window; POM-H Delrin parts | Laboratory tests of HCHO emissions (Section 4) |
| 108 | ECCC | 2016 | ARC window; POM-H Delrin parts | Field measurements (Section 2) |
| 118 | KNMI | 2016 | ARC window; POM-H Delrin parts | Laboratory tests of HCHO emissions (Section 4) |
| 148 | Virginia Tech | 2018 | temperature sensor (April 2019), wedged window; POM-H Delrin parts | Temperature (Section 3); Laboratory tests of HCHO emissions (Section 4); Field study (MAX-DOAS, Section 5.2) |
| 155 | Boston University | 2019 | temperature sensor; wedged window; POM-H Delrin parts | Temperature (Section 3) |
| 165 | EPA | summer 2019 | Nylon parts; temperature sensor; wedged window | Laboratory tests of HCHO emissions (Section 4) |
| 167 | EPA | summer 2019 | Nylon parts; temperature sensor; wedged window | Laboratory tests of HCHO emissions (Section 4) |
| 168 | EPA | summer 2019 | Nylon parts; temperature sensor; wedged window | Laboratory tests of HCHO emissions (Section 4) |

ECCC: Environment and Climate Change Canada; KNMI: Royal Netherlands Meteorological Institute; EPA: US Environmental Protection Agency; NASA: US National Aeronautics and Space Administration

**Table 2.** History of Pandora hardware changes related to direct sun HCHO measurements

| Period | Hardware components | Impact on HCHO | HCHO Data Used |
|---|---|---|---|
| 2007 - winter 2016 | parallel window, POM-H Delrin | window caused etalloning in direct sun measurements, HCHO emissions from POM-H Delrin - **direct sun HCHO is not correctable** | MAX-DOAS: Pinardi et al., 2013; Direct Sun: Park et. al., 2018 |
| spring 2016 - 2017 | anti reflective coating on parallel widow, POM-H Delrin | ARC degrades within 1 year of installation, temperature dependent HCHO internal emission from POM-H Delrin (disagreement between direct sun total column and MAX-DOAS tropospheric column), can be corrected for functioning ARC | MAX-DOAS: Kreher et al., 2020; Direct Sun: Spinei et al., 2018, Herman et al., 2018; Spinei et al., 2020 |
| 2018 - spring 2019 | wedged window*, POM-H Delrin | temperature dependent HCHO internal emission from POM-H Delrin (disagreement between direct sun total column and MAX-DOAS tropospheric column), can be corrected | MAX-DOAS: Nowak et al. (2020) |
| summer 2019- | wedged widow, nylon | believed not to have any interference caused by design (confirmed by extensive laboratory studies) | |

Note: HCHO from direct sun is not a standard PGN product and was not provided by the NASA and Luftblick PGN groups outside of KORUS-AQ study (Spinei et al., 2018, Herman et al., 2018). Park et al., 2018 performed HCHO analysis independently and were not aware of any PGN discoveries.
* wedged window are installed on new instruments, if the instruments were not returned to NASA or SciGlob - they are not upgraded, therefore some instruments are probably still have degrading ARC windows

---

## Author Response (AR1)

**Letter to Editor**

Dear Dr. Stutz,

We thank you very much for your consideration of our article titled: "Effect of Polyoxymethylene (POM-H Delrin) offgassing within Pandora head sensor on direct sun and multi-axis formaldehyde column measurements in 2016 - 2019" for pub-

5 lication in AMT. We found the reviewers' comments and suggestions very valuable and helpful in improving our manuscript. We have made revisions according to the reviewers' comments and suggestions, as described below. Reviewers' comments are given in regular font, authors responses are in italics font. We are also attaching a marked up version showing the changes.

**Best Regards, Elena Spinei and co-authors.**

10

15

**Responses to comments by Anonymous Referee 1**

**0.0.1 General comments:**

I would recommend adding one or two tables summarizing 1) which instruments have been used when/for which part of this work, to help the reader following which instruments have been participating to several steps;

We added table 1 describing Pandoras used in this study with their appropriate modifications and part of this study where they have been used.

| Ν   | Owner             | Manufactured | Relevant Hardware Components                                   | Contribution to This Study            |
|-----|-------------------|--------------|-----------------------------------------------------------------------|---------------------------------------|
| 2   | NASA              | 2011         | upgrade in summer 2019: Nylon parts,                                  | Temperature (Section 3); Field study  |
|     |                   |              | temperature sensor, wedged window                                     | (direct sun, Section 5.1)             |
| 21  | NASA              | 2011         | upgrade in 2016: ARC window;                                          | Laboratory tests of HCHO emissions    |
|     |                   |              | POM-H Delrin parts                                                    | (Section 4)                           |
| 32  | NASA              | 2016         | ARC window; POM-H Delrin parts                                        | Field study (direct sun, Section 5.1) |
| 46  | NASA              | 2015         | upgrade in 2016 ARC window;                                           | Laboratory tests of HCHO emissions    |
|     |                   |              | POM-H Delrin parts                                                    | (Section 4)                           |
| 108 | ECCC              | 2016         | ARC window; POM-H Delrin parts                                        | Field measurements (Section 2)        |
| 118 | KNMI              | 2016         | ARC window; POM-H Delrin parts                                        | Laboratory tests of HCHO emissions    |
|     |                   |              |                                                                       | (Section 4)                           |
| 148 | Virginia Tech     | 2018         | temperature sensor (April 2019),
wedged window; POM-H Delrin parts | Temperature (Section 3); Laboratory   |
|     |                   |              |                                                                       | tests of HCHO emissions (Section 4);  |
|     |                   |              |                                                                       | Field study (MAX-DOAS, Section 5.2)   |
| 155 | Boston University | 2019         | temperature sensor; wedged window;                                    | Temperature (Section 3)               |
|     |                   |              | POM-H Delrin parts                                                    | Temperature (Section 5)               |
| 165 | EPA               | summer 2019  | Nylon parts; temperature sensor;                                      | Laboratory tests of HCHO emissions    |
|     |                   |              | wedged window                                                         | (Section 4)                           |
| 167 | EPA               | summer 2019  | Nylon parts; temperature sensor;                                      | Laboratory tests of HCHO emissions    |
|     |                   |              | wedged window                                                         | (Section 4)                           |
| 168 | EPA               | summer 2019  | Nylon parts; temperature sensor;                                      | Laboratory tests of HCHO emissions    |
|     |                   |              | wedged window                                                         | (Section 4)                           |

**Table 1. Pandora instruments used in this study**

ECCC: Environment and Climate Change Canada; KNMI: Royal Netherlands Meteorological Institute; EPA: US Environmental Protection Agency; NASA: US National Aeronautics and Space Administration

2) the different steps of the Pandora HCHO "history" with some links to published HCHO Pandora papers: - pre2016: no HCHO possible; - spring 2016: window replacement and KORUS-AQ and CINDI-2 but issue of coherence between MAX-DOAS and direct-sun results; - april 2019: added temperature sensors for a few instruments and highlight of the potential link

of HCHO creation with temperature; - lab measurements to confirm the findings; - summer 2019: new head sensor without 20 POM-H Delrin to solve the problem; - measurements campaigns illustrating the findings.

We added table 2 describing Pandora design modifications including references

| Period             | Hardware components                                              | Impact on HCHO                                                                                                                                                                                                                               | HCHO Data Used                                                                                                    |
|--------------------|------------------------------------------------------------------|----------------------------------------------------------------------------------------------------------------------------------------------------------------------------------------------------------------------------------------------|-------------------------------------------------------------------------------------------------------------------|
| 2007 - winter 2016 | parallel window,
POM-H Delrin                                 | window caused etalloning in direct sun
measurements, HCHO emissions from POM-H
Delrin - direct sun HCHO is not correctable                                                                                                      | MAX-DOAS: Pinardi et al., 2013;
Direct Sun: Park et. al., 2018                                                 |
| spring 2016 - 2017 | anti reflective
coating on
parallel widow,
POM-H Delrin | ARC degrades within 1 year of installation,
temperature dependent HCHO internal emission
from POM-H Delrin (disagreement between direct
sun total column and MAX-DOAS tropospheric
column), can be corrected for functioning ARC | MAX-DOAS: Kreher et al., 2020;
Direct Sun: Spinei et al., 2018,
Herman et al., 2018; Spinei et al.,
2020 |
| 2018 - spring 2019 | wedged
window*,
POM-H Delrin                               | temperature dependent HCHO internal emission
from POM-H Delrin (disagreement between direct
sun total column and MAX-DOAS tropospheric
column), can be corrected                                                                    | MAX-DOAS: Nowak et al. (2020)                                                                                     |
| summer 2019-       | wedged widow,
nylon                                           | believed not to have any interference caused by
design (confirmed by extensive laboratory studies)                                                                                                                                        |                                                                                                                   |

 Table 2. History of Pandora hardware changes related to direct sun HCHO measurements

Note: HCHO from direct sun is not a standard PGN product and was not provided by the NASA and Luftblick PGN groups outside of KORUS-AQ study (Spinei et al., 2018, Herman et al., 2018). Park et al., 2018 performed HCHO analysis independently and were not aware of any PGN discoveries.

\* wedged windows are installed on new instruments, if the instruments are not returned to NASA or SciGlob - they are not upgraded, therefore some instruments probably still have degrading ARC windows

In the light of the "pre-2016" first point, there should also be some comments of Pandora HCHO measurements performed before 2016 (e.g., in Pinardi et al. 2013, for measurements in 2009 – is there any other ?).

Table 2 contains periods of operation and the publications used data from that period to the best of our knowledge

I would also include some more references in the introduction on satellite HCHO (only one is mentioned, while at least 8 are known - and given below) and on satellite validation.

**We added the recommended references**

In the abstract and/or conclusions, it would also be nice to also convert the findings from DU to molec/cm2, as this unit is usually used for HCHO retrievals.

30 We added "2.69 molecules/ $cm^2$ " to the abstract. There is a definition of DU in text p.2, line 42

**0.0.2 Specific comments and technical corrections:**

- P2, line 29-33: give some references for the other satellites, as done for SCIAMACHY. Several are suggested below.

We added the recommended references

- P2, line 37: give more references for other validation studies. Several are suggested below.

We added the recommended references

35

P2, last line: Lamsal et al. 2014 and Kollonige et al. 2017 are not given in the reference list. Consider adding other recent works with several Pandora NO2 measurements used for validation (Herman et al., 2019; Pinardi et al., 2020; Verhoelst et al., 2020).

**- We added the recommended references**

40 Please check completeness of the reference list in the whole manuscript (e.g., Reed et al., 2015, Gronoff et al., 2019 are also missing in the references).

We have checked the references

- P3, line 65 to 71: it would be nice to link the different steps, with the manuscript sections. Or at least add the link with the sections in the proposed additional table n2.

**45 We added reference to the corresponding sections**

- P3, line 74: Retrieval of weak absorbers such as HCHO was not possible from 'the pre-2016 Pandora direct sun measurements due to the telescope assembly front window etaloning'': comment Pinardi et al., 2013 measurements of 2009.

Pinardi et al. 2013 reference MAX-DOAS HCHO only, no HCHO direct sun data from Pandora are presented. Only coherent light (e.g direct solar beam) causes etalloning by the window.

50 - P4, figure 1: please revisit the legend. What are the light blue points?

Both blue and light blue points represent HCHO emissions from the same instrument P46 but at different temperature ramp rates. We believe this is appropriate.

- P4, line 95: . . . the reference spectrum is \*often\* taken within. . . (not for the CINDI-2, as mentioned later in the paper)

MAX-DOAS data analysis for profiles is done with the scan reference. CINDI-2 MAX-DOAS data for profiles was also analysed with the scan reference (Tirpitz et al., 2020). Local noon reference was used for  $\Delta S$  intercomparison not the final inversion. Zenith  $\Delta S$  were subtracted from the  $\Delta S$  at all other elevation angles

- P5, line 108: "A baffle holding tube, two filter wheels, and a dark filter" -> The baffle holding tube, the two filter wheels, and the dark filter. . .

corrected

60 - P7, line 149: remove parenthesis in "Figure (4). . ."

corrected

- P7, line 153: consider changing "follow-up work" to the relevant section of the manuscript.

We removed this sentence

- P7, line 160: add a comma before 118

65 *corrected*

55

- P7, lines 147 to 148 and lines 160 to 161: as proposed in the general comments, consider adding a table with the Pandora numbers, their construction year, and in what part of this work they appear.

Table 1 was added

- P8, line 161: "Three other Pandora" -> The other three Pandora

**70 *corrected**

- P8, line 169: consider adding here the information on the 1000 W FED lamp given in line 186 and the LED lamp (cf line 2018 and 2016), or removing this information from line 169.

**We removed this sentence**

- P8, line 170: what is the meaning/purpose of the "The head sensor was not disturbed during the measurements" ?

75

We intended to emphasize that the instrument position was not changed. We removed it.

- P8, line 175 to 179: these DOAS settings seems the generic ones used in the laboratory study (332-360nm), but actually in P10 line 250, these are different (300-350nm) (and also different than the in-field settings, P13 line 396, 332-359nm), so this is a bit perturbing. Is there any reason why changing wavelength ranges, polynomial and offset order? For the inclusion of trace gases, this is clear/mentioned when relevant.

80 We performed identical tests for all pandora instruments. The new pandoras had no detectable HCHO absorption in multiple fitting windows. Since the results were not impacted by the window choice one single fitting window was chosen for both SO2 and HCHO retrieval. We also wanted to emphasise the low fitting residual for 50 nm fitting window.

P9, line 193: "using the pre-installed Bosch BME280 digital humidity, pressure and temperature sensor on Spark-Fun Atmospheric Sensor Breakout Board" -> this is a detail of the new head sensor. It would be maybe better to introduce it in Sect
 2.3 when introducing the Pandora internal head sensor temperature?

We added the information about the internal temperature sensor to Section 2.3

- P9, lines 208 to 210: the Pandora 118 was also tested with a LED lamp in addition to the FEL lamp. Is this shown somewhere? Why is this done? What was expected (differently) from this additional test?

Some literature suggested that UV and VIS radiation (200-800 nm) was responsible for Photo-oxidation at those wavelengths. We were curious whether more HCHO will be emitted while using FEL vs LED light source. We did not see any statistically meaningful difference between the retrievals.

- P10 and P11 (figure 6): POM-free and Delrin-free. These are used as synonyms? In P13, line 281, also POM-H is used. Use one name everywhere.

We corrected all to POM-H Delrin free

95 - P12, line 268 and elsewhere in the manuscript: replace Kreher et al., 2019 by Kreher et al., 2020.

corrected

- P12, line 269: there are two point at the end of the line.

corrected

- P14, line 314: "contribution" seems to be in bold
- 100 *corrected*
  - P17, line 348: add reference to Fig 5 for the "small generation rates inside Pandora 148 head sensor (a = 0.0041 DU)"?

corrected

- P17, lines 53: give an estimation and a reference (Pinardi et al., 2013 ?) of the "fitting noise for most DOAS instruments" (usually in molec/cm2!).

105 We added:  $(0.3 \text{ DU} = 8 \times 10^{15} \text{ molecules/cm}^2$ , Table 7 in Kreher et al., 2020.

- P17, line 357: add "Sept. 2016" after the coordinates and update Kreher reference.

added

- P18, line 378: if I followed well all the Pandora numbers, 32 was involved in the GSFC field campaign, and 31 and 46 in the lab measurements wrt temperature. A table would definitely help follow which instrument has been used when!

110 Table 1 was added

- P20: suggestion to add in each bullet conclusions the Pandora numbers relevant to each bullet (or not if it is already clear in an additional table), and the relevant Figure supporting each conclusion (e.g., Fig 5 for the 1rst point). Also add molec/cm2 estimation in addition to DU values.

Since Table 1 was added we have not modified the Pandora listing in conclusions. We added Figure references to the conclusions.

- P20, point 5: maybe cite publications that were made with Pandora direct sun HCHO data between 2016 and 2019, and that should not be considered "valid"?

We added references to those publications in Table 2. We replaced "build" with "operational" to emphasise that any other Pandora data whether part of a field campaign or routine measurements cannot be used in the current form

120

**Response to comments by Anonymous Referee 2**

**0.0.3 General comments:**

The manuscript provides a detailed timelines and descriptions for multiple events such as Pandora sensor head design changes; laboratory testing for POM and non-POM units; and case-studies for real-life deployments/colocations. I found myself flipping back and fourth through the manuscript, I therefore think that the manuscript will be improved by inclusion of a master table summarizing the types of tests, dates, intercomparison campaigns, identification numbers of units, modifications to the units,

125

130

etc. This table can be placed at the end of Section 1.

We have added Table 1 (see the attached manuscript) that summarizes the Pandora instrument information and how they contributed to this study. We also added the following sentence at the end of Section 1: "Table 1 lists the Pandora instruments description and contribution to this study.". We did not add the actual field campaigns that each pandora participated in to make sure we do not give a wrong impression that only these instruments are impacted.

Can authors develop a correction factor that can be applied for direct sun HCHO data collected during 2016-2019 to correct for HCHO production, so the dataset can be utilized by the scientific community? For example, recommending temperature ranges during which data would be usable, and showing examples of intercomparison with in-situ techniques (if available) or with satellite data showing a reasonable agreement.

We have performed detailed evaluation of Pandora head sensor heat transfer and estimation of HCHO production amplitude. We are preparing a separate manuscript: Spinei et al. 2020 (in preparation). We added the following reference to point 5 of the conclusions: " Considering that Pandora head sensors have almost identical design from material, shape and thermodynamics point of view data between 2016 and 2019 can be corrected based on (a) meteorological data (temperature and wind) to estimate internal head sensor temperature and (b) on ΔS measurements to estimate 140 HCHO production amplitude (Spinei et al. 2020 in preparation)"

**0.0.4 Specific comments:**

Lines 10-14: define cold and warm temperature ranges. Remove quotation marks from "cold"

We added the temperature information: Measurements in winter, during colder (<10°C) days in general and at high solar zenith angles (> 75 °) were minimally impacted. Measurements during hot days (>28°C)

145 Figure 1: define light blue, gray and green lines

We added the following information to the figure caption: (green: box with a median mixing layer height (MLH), grey: box with a measured MLH; light blue: box+exponential profile with a median MLH, and black: box+exponential profile with a measured MLH,

Lines 91-98: add explanation for which spectra are used for direct sun and multi-axis DOAS retrievals.

150 We have made the following modification: DOAS implementation of multi-axis retrieval is significantly less sensitive to instrumental changes. This is due to the fact that single scan sky scattered solar spectra are analysed using a zenith reference spectrum taken within maximum 10-15 minutes from the scan measurements. Direct sun spectra, on the other hand, are analyzed using a single reference spectrum that was potentially taken months apart from the rest of the spectra.

Figure 2: add ambient temperature to the figure

155 added

Line 160: add a coma before 118

added

Line 161: add a coma before and 148

added

160 Line 171: remove quotation marks from, "mimicking"

removed

Line 314: remove bold face from contribution

removed

Lines 404-405: The statement "Pandora HCHO measurements derived from direct sun observations between 2016 and 2019
 165 cannot be used in the current form. Results presented here most likely are representative of other instruments build between 2016 and 2019" is very drastic. Authors should consider adding recommendations on possible corrective approaches, so the data could be utilized by scientific community.

We are preparing a publication to describe such corrections. We added: "Considering that Pandora head sensors have almost identical design from material, shape and thermodynamics point of view measurements between 2016 and 2019
 170 can be corrected based on (a) meteorological observations (temperature and wind) to estimate internal head sensor temperature and (b) on ΔS measurements to estimate HCHO production amplitude (Spinei et al. 2020 in preparation)

**Effect of Polyoxymethylene (POM-H Delrin) offgassing within Pandora head sensor on direct sun and multi-axis formaldehyde column measurements in 2016 - 2019**

Elena Spinei1, Martin Tiefengraber2,3, Moritz Müller2,3, Manuel Gebetsberger2, Alexander Cede2, Luke Valin4, James Szykman4, Andrew Whitehill4, Alexander Kostakis5, Fernando Santos6, Nader Abbuhasan7, Xiaoyi Zhao8, Vitali Fioletov8, Sum Chi Lee8, and Robert Swap9

[revised manuscript text omitted]